# Structural conversion of the spidroin C-terminal domain during assembly of spider silk fibers

Danilo Hirabae De Oliveira[1,2], Vasantha Gowda[2], Tobias Sparrman[3], Linnea Gustafsson [4], Rodrigo Sanches Pires [2], Christian Riekel [5], Andreas Barth [6], Christofer Lendel [2] ✉ & My Hedhammar [1] ✉

The major ampullate Spidroin 1 (MaSp1) is the main protein of the dragline spider silk. The C-terminal (CT) domain of MaSp1 is crucial for the self-assembly into fibers but the details of how it contributes to the fiber formation remain unsolved. Here we exploit the fact that the CT domain can form silk-like fibers by itself to gain knowledge about this transition. Structural investigations of fibers from recombinantly produced CT domain from *E. australis* MaSp1 reveal an α-helix to β-sheet transition upon fiber formation and highlight the helix Nº4 segment as most likely to initiate the structural conversion. This prediction is corroborated by the finding that a peptide corresponding to helix Nº4 has the ability of pH-induced conversion into β-sheets and self-assembly into nanofibrils. Our results provide structural information about the CT domain in fiber form and clues about its role in triggering the structural conversion of spidroins during fiber assembly.

Web-spinning spiders are specialized arachnids that produce several types of silks used to capture prey, frame the egg case, escape from predators, attach to surfaces, and others. The dragline silk is a robust thread, elastic and strong enough to absorb and dissipate high energy loads[1]. In addition, it has exceptional qualities for a biomaterial, being naturally degradable, compatible with cell growth, and not provoking immune response[2]. Accordingly, spider dragline silk has been suggested for use in both medical and industrial applications[3,4].

The mechanical properties of spider silk are an innate consequence of the advanced structural organization of proteins known as spidroins. The major ampullate spidroins (MaSp) 1 and 2 are the main element of the dragline silk. These large proteins (size ranging from 200 kDa to 350 kDa) have specific arrangements, with a central repetitive region allocated between non-repetitive and evolutionary conserved N- and C- terminal (NT and CT) domains (Fig. 1a). Due to the complexity of natural silk fibers and the considerable size of the full-length spidroins, it has been challenging to elucidate the structure of native fibers. Lately, the recombinant production of partial spidroins has provided clues about the macromolecular structure of silk proteins in solution and fibers. Both CT and NT domains have been reported to be important for triggering fiber formation[5-7]. NT has been shown to increase the spidroin solubility and to control fiber formation of spidroins through a pH-dependent dimerization[5,8]. Several strategies for fiber formation of partial recombinant spider silk proteins with or without the terminal domains have been explored, including the ADF3/ADF4 from *A. diadematus*[9], and the consensus repetitive region from *N. clavipes*[10], among others[11]. A chimeric version with the repetitive segment and NT from *E. australis*, and CT from *A. ventricosus* minor ampullate (MiSp), forming NT2RepCT, has shown good production yield[12]. Diverse techniques have been applied to convert the soluble recombinantly produced spidroins into fiber-like structures, (ranging from micrometers to nanometers in diameters and variated

[1]Department of Protein Science, School of Engineering Sciences in Chemistry, Biotechnology and Health, KTH Royal Institute of Technology, AlbaNova University Center, Stockholm, Sweden. [2]Department of Chemistry, KTH Royal Institute of Technology, Stockholm, Sweden. [3]Department of Chemistry, Umeå University, Umeå, Sweden. [4]Spiber Technologies AB, Roslagstullsbacken 15, 114 21 Stockholm, Sweden. [5]European Synchrotron Radiation Facility, B.P. 220, F-38043 Grenoble Cedex, France. [6]Department of Biochemistry and Biophysics, Stockholm University, Stockholm, Sweden. ✉e-mail: lendel@kth.se; myh@kth.se

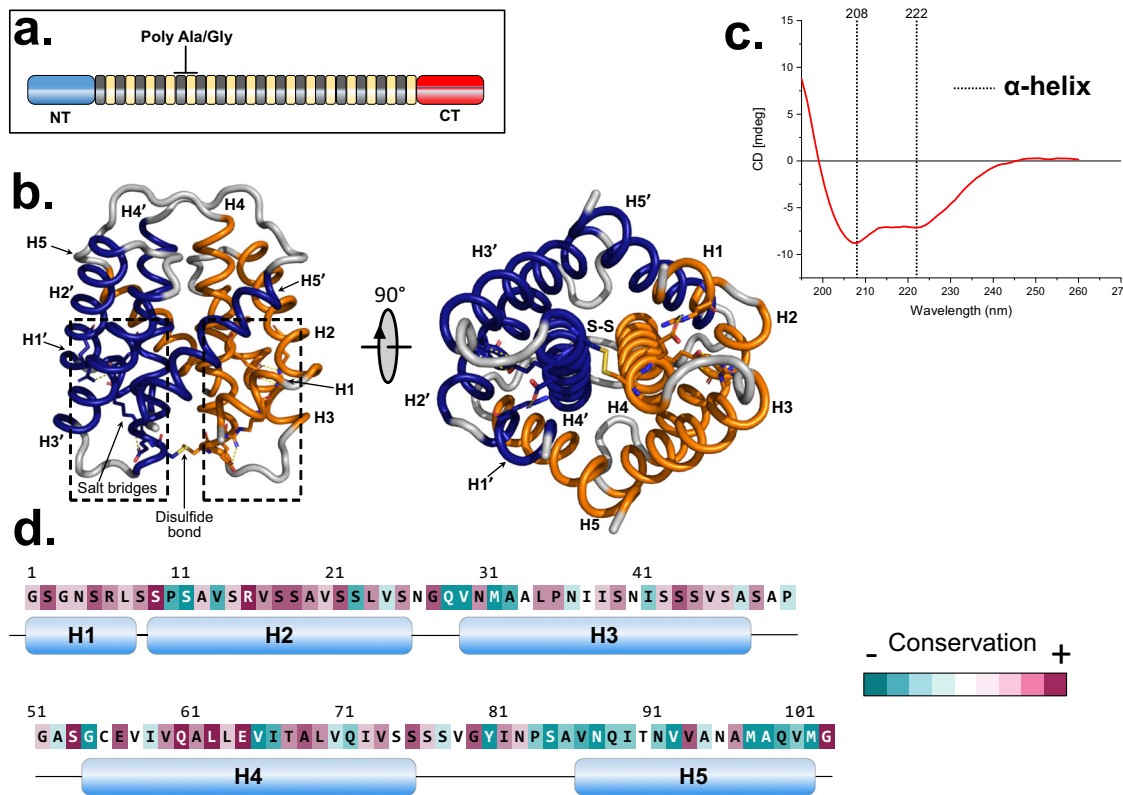

**Fig. 1 | Overview of spidroin and CT domain (*E. Australis*) structure. a** Schematic illustration of full-length major ampullate spidroins, including: N-terminal domain (NT), repetitive central region (poly-Ala/Gly), and C-terminal domain (CT). **b** The comparative modeling (homology model) of CT domain of *E. australis* MaSp1[86], shown with chain A in blue, chain B in orange and the inter-chain disulfide bridge represented in yellow. The amino acid residues forming salt bridges are visualized with stick representation where the negative charges are red and the positive charges are blue. **c** CD spectroscopy of CT domain (MaSp1 *E. australis*) at pH 8 presents the canonical α-helix pattern with minima at 222 nm and 208 nm. **d** The sequence of CT with the α-helices indicated, and the conservation score according to ConSurf [83]. The images were rendered using Pymol.

length), including wet spinning into coagulation baths[13], electrospinning[14,15], and biomimetic spinning using microfluidic devices[12].

The recombinantly produced partial spidroin 4RepCT, consisting of four repetitive poly-Ala/Gly rich stretches and the CT domain from MaSp 1 of *E. australis*, self-assembles at air-water interfaces, i.e., without any denaturing agents or heat, into fibers resembling native spider silk, and has therefore been used as a model for structural studies[6]. The CT domain by itself can also self-assemble into fibers under similar conditions, although with lower stability (they can be dissolved in urea, which is not the case for 4RepCT fibers)[6,7,16]. The NT domain cannot self-assemble into fibers using the same set of conditions, neither on its own nor followed by a part of the repetitive region (without CT domain)[5].

The CT domain is a non-repetitive and highly conserved protein composed of ~100 amino acids. It has been postulated that the CT domain may trigger the fiber assembly by initiating the alignment of the repetitive domains[6]. Hence, elucidation of the process by which the CT domain forms fibers could improve the understanding of structural transitions also in other silk constructs. The homologous structure of a recombinantly produced CT domain of dragline silk protein ADF-3 from *A. diadematus* in solution has been experimentally determined by nuclear magnetic resonance spectroscopy (NMR)[17]. The soluble state is an α-helical homo-dimer in which the chains are connected by an interchain disulfide bond in helix Nº4 of each chain. The stability of the CT domain (from *A. diadematus* and most likely also from *E. australis* MaSp1) in solution is further increased by a pair

of intramolecular salt bridges, connecting helix Nº1 and helix Nº2 to the same side of helix Nº4.

The CT domain has been shown to be present in spider's native dragline silk threads post-spinning[18]. Previous attempts to characterize a recombinant CT domain in various pH solutions have given a glimpse into its behavior in the fiber[19]. However, participation and folding of the CT domain in the fiber form remain poorly understood[20]. While full-length spidroins are amphiphilic and have key hydrophobic clusters composed of poly-Ala stretches in the repetitive region, the CT domain has a high degree of conserved residues but no tandem Ala motifs[21]. Moreover, the CT domain contains the most hydrophobic region of the entire spidroin (in case of *E. australis* MaSp1 at helix Nº4: ALLEVITALVQIV), which could be essential for protein folding in solution and liquid-air interface adsorption during the transition to the fiber state[22].

Previous studies of native silk using solid state NMR (ssNMR) spectroscopy[23], Fourier transform infrared (FTIR) spectroscopy[24], Raman spectroscopy[25], nanofocusing X-ray scattering[26], and molecular dynamics simulations[27] have mainly focused on the structure of the repetitive region. They have shown that the poly-Ala stretches form β-nanocrystalline structures. Parallel investigations using those solid-state methods have investigated the atomistic features of secondary and higher-order structures of recombinant spider silk[28,29] and fibrous proteins in general[30]. As a result, the solid-state techniques have emerged as essential tools to characterize macromolecular complexes that cannot be crystallized but are obtained in other solid forms. Here we use a range of solid-state methods to elucidate the structural

characteristics of the CT domain in fiber form assembled at the air-liquid interface. Together with computational modeling and predictions, this allows us to suggest a model for how the CT domain could initiate silk assembly.

## Results

### Comparative modeling and circular dichroism spectroscopy of the CT domain in solution present α-helical content

There is no experimentally determined high-resolution molecular structure of the herein-used CT domain from *E. australis* in solution. Based on the sequence similarities, a model was created using comparative modeling and the structure of the CT domain from ADF-3 of *A. diadematus* (PDB ID: 2KHM) as template. The thereby obtained homology model suggests that CT domain from *E. australis* is compatible with a dimer of five-helix bundles with tight interchain interactions, including a disulfide bond (Fig. 1b). A predominantly α-helical structure of the soluble CT domain is also in agreement with circular dichroism (CD) spectroscopy data (Fig. 1c, Table S1 and ref. 7)[31]. CD spectroscopy analyses show that the α-helical structure of the CT domain is initially kept at lower pH conditions, but the signal decreases over time, due loss of soluble protein (Fig. S1).

One interesting feature of the CT domain structure is that the disulfide link aligns helices Nº4 in the two protein chains in a parallel manner, coexisting with one of the most conserved hydrophobic regions (Fig. 1d and Fig. S2), which could potentially be a nucleation point for a structural conversion to parallel β-sheets.

The intermolecular disulfide bond of the CT domain is likely increasing its chemical and thermal stabilities[22]. In the presence of the reducing agent dithiothreitol (DTT), the soluble and denatured form of the CT domain moves as one, monomeric band (~10 kDa) upon SDS-PAGE analysis, while in the absence of DTT, a second band (~20 kDa) is also visible (Fig. S3), suggesting that a fraction of the protein is disulfide-linked in solution. Moreover, fibers formed from the same CT domain and then dissolved with 6 M urea in absence of reducing chemicals presents one band around 20 kDa (Fig. S4), indicating that the CT domain in fiber form is a disulfide linked dimer. Notably, stable dimers have been reported also for CT domain lacking the disulfide bond[32,33].

### FTIR spectroscopy reveals β-sheet content of the CT domain in fiber form

The recombinantly produced CT domain was transformed into a fiber form (Fig. 2a and Fig. S5) using self-assembly at the air-liquid interface with the stretch-relaxation cycles method[6,7,34]. With this method, fiber self-assembly of the CT-domain is efficient under native-like conditions (tris buffer pH 8.0), likely due to that the process includes shear-stress of the air-water interface.

To investigate the secondary structure of CT in fiber form, the FTIR spectrum of the amide I region (1600–1700 cm⁻¹)[35,36] was measured using Attenuated Total Reflection (ATR) (Fig. 2b, c). The β-sheet content of the CT fiber can be seen in the absorbance spectrum from the clear and strong shoulder in the 1630–1610 cm⁻¹ range as well as from a minor component band at 1696 cm⁻¹, often assigned to antiparallel β-sheets[37,38]. However, another peak is found at 1650 cm⁻¹, showing that also α-helix structures are present in the CT fiber. In total, the obtained spectrum suggests a secondary structure content of ~45% β-sheet, ~40% α-helix, and ~16% of other structures[38,39] (Table S2a).

In order to complement the structural analysis of CT fibers with an understanding of the CT domain in solution and its secondary structure prior to self-assembly, a drop of CT solution was placed on the diamond surface of the ATR unit and left to air dry, and then analyzed using FTIR spectroscopy (Fig. S6). The amide I band of this sample suggests a clear difference in the secondary structure content compared to the same protein processed into a fiber using the stretch-

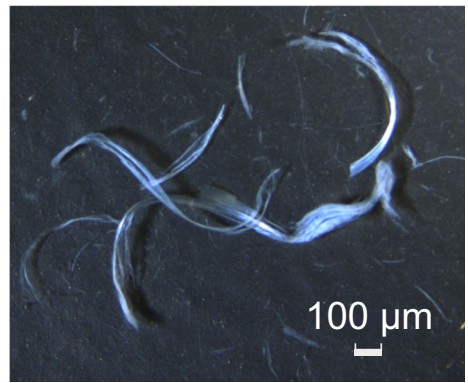

a.

100 µm

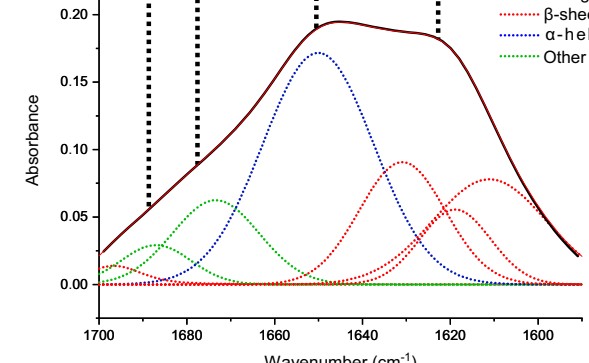

**Fig. 2 | FTIR spectra of CT fiber in the amide I region. a** Fibers of the recombinantly produced CT domain obtained by a stress-relaxation cycle method. **b** Second derivative of the absorbance spectrum (black) and the fit (red) for the CT fiber. **c** Absorbance spectrum of the CT fiber (black), the fit (red) with the same fit model as in (**b**), and the fitted component bands (dotted lines)[72]. Other secondary structures refers to irregular, turns, bends, and other helix types.

relaxation cycles method. For the dried drop, the content of α-helix is 35%, of β-sheet 32%, and of other structures 31% (Table S2b). When compared, the β-sheet content of the CT fiber is thus increased, which can be seen in the shoulder in the absorbance spectrum from 1630–1610 cm⁻¹ range. Besides, the CT fiber sample is characterized by a component band at 1696 cm⁻¹ (anti-parallel β-sheets), which is considerably reduced for the dried drop of CT. A band found at 1650 cm⁻¹,

assigned as α-helix, is found in both samples, suggesting that the CT domain may share similarities regarding the structure of α-helices in both soluble and fiber forms.

To estimate the relative proportions of parallel and anti-parallel β-sheets, we calculated the β-sheet organizational index[40]. The index was 0.1 for the CT fiber, which is between those for proteins rich in anti-parallel β-sheets (0.2–0.3) and for proteins rich in parallel β-sheets ($\leq$0.05)[40], thus suggesting a mixture of both β-sheet types in the CT fiber, with some prevalence of parallel conformation (Fig. S7).

## CT fibers show a distinct X-ray diffraction pattern

Wide-angle X-ray scattering (WAXS) can be used to provide information about interatomic distances and is frequently used to characterize the structure of fibrous proteins[41]. Herein, we compared fibers of the CT domain only, with fibers of the recombinant partial spidroin 4RepCT, consisting of four repetitive poly-Ala/Gly-rich regions and the same CT domain from MaSp1 of *E. australis*[7]. The homogeneity of the samples was checked by raster-scans. The fiber samples appear to be homogeneous and without apparent skin/core morphology (Fig. S8a). The overall X-ray diffraction pattern of 4RepCT fibers (Fig. 3b, d) resembles those of natural spider silk, but with less order[42]. A broad diffuse background, due to a short-range order fraction, and Bragg reflections, due to a crystalline fraction, can be seen. When measuring dragline silk obtained from spiders, the diffraction pattern typically shows β-poly(L-Ala) nanocrystallites that are highly oriented along the fiber axis[43–45]. The X-ray diffraction of fibers from recombinantly produced 4RepCT presents reflection rings with moderately arced configuration denoting a partially oriented crystalline fraction, with a strong inter-strand distance of 4.4 Å (slight meridional arc). Reflections corresponding to 3.74 Å, originating from intra-chain Cα repetitions[46], can also be seen. A weaker equatorial arc corresponding to inter-sheet distance of 5.1 Å, and a weaker reflection ring corresponding to 9.67 Å, is also seen from the 4RepCT fiber. The inter-sheet spacing is generally determined by the length of the sidechains[47,48]. Tightly, complementary interfaces of repetitive Ala side chains contribute to shorter inter-sheet distances, and the diffractions of 5.1 Å (Table S3) agree very well with those of β-poly(L-Ala)[47].

In the pattern of CT fibers, the diffraction peaks related to the hydrogen-bonded spacing between the β-strands is slightly shifted to 4.59 Å, and the reflections representing the periodic (Cα) organization along the chain direction appear at 3.88 Å. Notably, as expected from the sequence, the inter-sheet distances typical for poly-Ala are absent in the diffraction pattern from CT fibers. Instead, the reflection at 9.75 Å becomes more prominent (Fig. 3a, c, Table S3), suggesting the presence of β-sheets formed by other residues with larger side chains than Ala. Such condition resembles the pattern of amyloid-like fibrils with inter-sheet spacing between 9-11 Å[49]. The amino acid content of the CT domain shows larger variation than for the sequence in the repetitive region. Thus, larger inter-strand distances were expected from the CT fiber, since its sequence contains β-branched side-chain and other bulky side-chains as the Glu.

The diffraction pattern of 4RepCT appears to be a convolution of the CT domain and β-poly(L-Ala) diffraction patterns, indicating that the crystallites are similar in fibers of CT domain and 4RepCT, and potentially also in natural silk. A third sample of a recombinant protein that only contains the 4Rep region of *E. australis* was examined to complement the X-ray analysis. Although the 4Rep protein cannot form coherent fibers, it forms aggregates and rudimentary fibrils[7] that were compatible with X-ray diffraction analysis. The obtained pattern of 4Rep aggregates show similarities to β-poly(L-Ala), thus in line with the convolution hypothesis (Fig. S8c and d).

The particle size values were estimated using Scherrer's method[50] and was found to be, ca. 40 Å for the 4RepCT fiber from the width of the 4.4 Å peak while the 9.7 Å peak provides a particle size of 23 Å (Fig. 3c). Similarly, for the CT fiber, the 4.65 Å peak provides a particle

size of ca. 50 Å. The degrees of crystallinity were estimated according to Eq. (1) and found to be $X_c = 21\%$ for the 4RepCT fiber and $X_c = 13\%$ for the CT fiber, which agrees to the less crystalline appearance of the CT pattern. The particle size and crystallinity index values derived for 4RepCT and CT fibers are thus in the range of values previously reported for major ampullate silk[26,51], where they are attributed to the dimensions of β-sheet nanodomains reinforcing a network of (ideally) random protein chains.

## CT fibers bind a molecular probe for amyloid detection

To further explore the presence of ordered β-sheet structures in the CT fiber, we investigated the binding to the dye Amytracker, developed for the detection of amyloid-like structures. Binding of the dye to repetitive arrangements of β-sheets, such as those found in amyloids, results in a distinct fluorescence signature of the dye. Fluorescence microscopy indeed confirms the binding of the dye to CT fibers (Fig. S9). Hence, the results support the hypothesis that parts of CT forms ordered β-sheet structures that have similarities to amyloid-like fibrils and silk nanocrystallites.

## Similarities between recombinant partial spidroins and natural spider silk observed by natural abundance [13]C NMR

The [13]C ssNMR spectra of natural silks obtained from diverse spiders are dominated by the poly-Ala and Gly portions of the spidroins (Fig. S10c)[52]. The chemical shifts of these peaks display subtle variations among the species, and the Ala Cβ shifts ≈21 ppm indicate substantial β-sheet content for all types of silk[53,54]. The spectrum of 4RepCT in fiber form is dominated by the high-intensity peaks at 49.3 ppm, 20.4 ppm, and 172.4 ppm corresponding to Ala Cα, Cβ, and C', respectively, suggesting a high content of β-sheet for Ala (Fig. S10b). The Cα/Cβ peaks of Ser are present in all types of silk, and the Cβ chemical shift (64.0 ppm) seems slightly shifted for 4RepCT compared to the natural silks. This could also be due to overlap with Pro Cα resonances. The natural spider silk is a mix of MaSp1 and MaSp2, the latter being proline-rich, which can be detectable in the spectra of natural silk around 60 ppm. Moreover, natural silk presents a significant Gly Cα component that is less dominant for 4RepCT. That is probably due to a higher relative content of Gly residues in natural silk, while 4RepCT has a higher Ala/Gly ratio than other silk types. Signals from aromatic residues (Tyr) are seen between 110 and 160 ppm in natural silk but are only weakly visible in 4RepCT (Tyr Cγ, Cδ, 130 ppm), likely due to a relatively low content in the 4RepCT sequence (Fig. S11).

Fibers made of only the CT domain were also analyzed using natural abundance [13]C ssNMR (Fig. S10a) and compared with 4RepCT fibers. The CP-MAS spectra of CT fibers present the highest intensity peak at 19.5 ppm, which may contain contributions from Ala Cβ but most likely also Cγ of Val. The relative Ala content in CT is lower than in 4RepCT, and the Ala Cα peak at 49.4 ppm has a lower intensity than in the spectrum of 4RepCT. A significant fraction of the Ala residues in 4RepCT occurs in the poly-Ala nanocrystalline structures presented in the repetitive part of 4RepCT and thus not present in the CT domain.

## 2D [13]C NMR suggests mixed secondary structure content in CT domain fibers

To get more detailed information about the structure of the CT domain in fiber form, we produced and analyzed fibers made of uniformly [13]C[15]N labeled CT protein. This strategy allows for the acquisition of 2D spectra as well as the investigation of [15]N chemical shifts. However, the [15]N spectra were found to be poorly resolved, with few peaks that have high enough intensity to allow assignment (Fig. S12). This indicates that the CT domain is most likely present as a mixture of different structural states and/or that not all the CT molecules have been efficiently transformed into fibers. Unfortunately, it also makes the sequential assignment of the protein difficult. On the

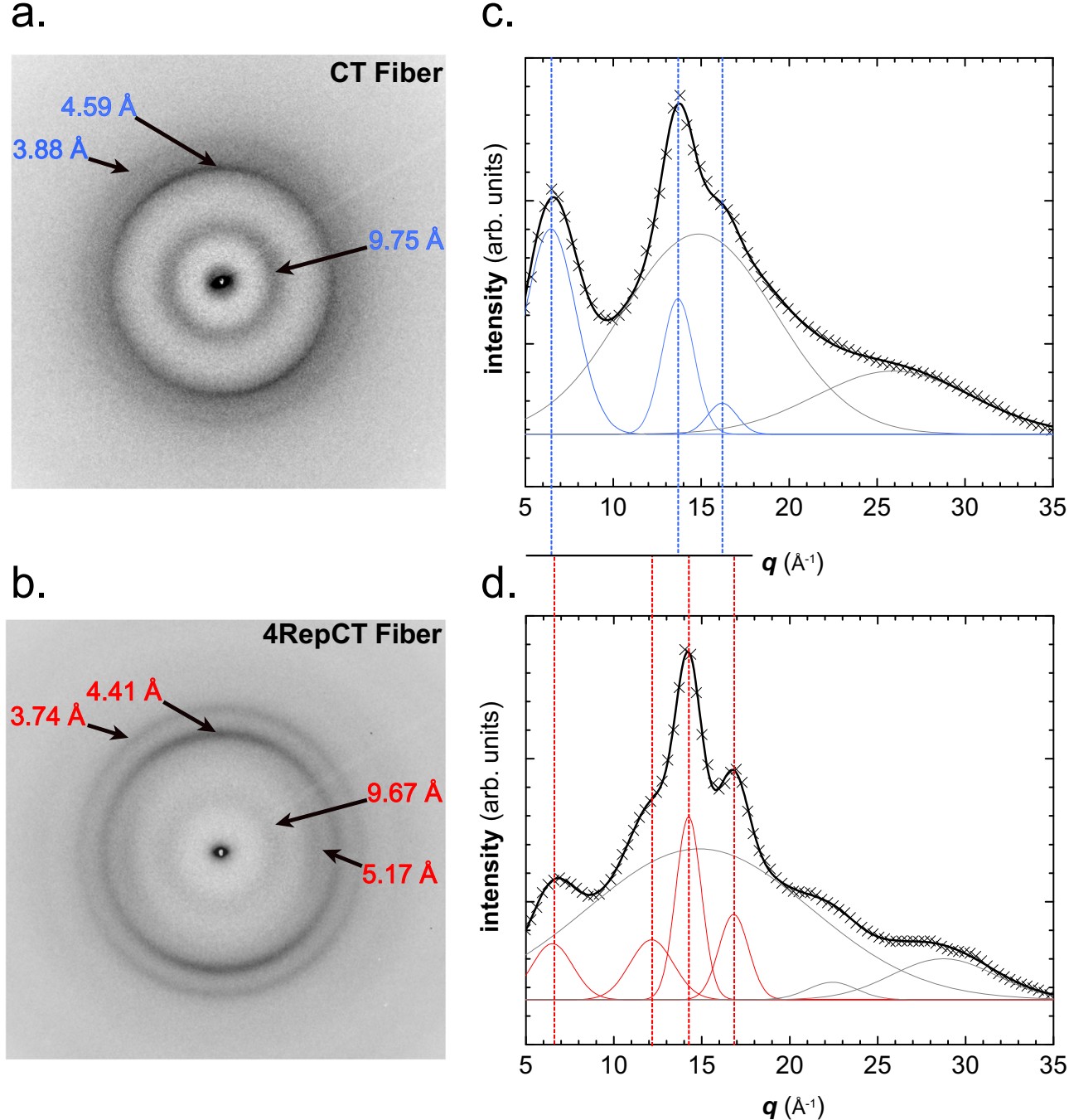

**Fig. 3 | The X-ray diffraction pattern of recombinant silk protein fibers.** Individual diffraction patterns (**a**, **b**) and azimuthally averaged pattern (**c**, **d**) of fibers from the CT domain (**a**, **c**) and the whole 4RepCT construct (**b**, **d**). The size of the side chains determines the packing of crystalline regions and thereby the inter-sheet distances.

other hand, cross-peaks for several types of amino acid residues could be identified in the $^{13}$C-$^{13}$C DARR correlation spectrum (Fig. 4, Tables S4 and S5). These peaks include Ala (α-helix, β-sheet, and coil), Val (α-helix, β-sheet, and coil), Ser (β-sheet), and Leu (α-helix).

Further analysis of the 2D $^{13}$C dipolar assisted rotational resonance (DARR) spectra with different mixing times allowed a more robust assignment of residues in β-sheet configuration (Asn, Gln, Glu, Gly, Ile) as well as with α-helical structure (Ser, Ile, Val, Leu, Met, Arg, and Asn). In addition, Ala and Val residues have minor peaks representing random coil structures (Fig. 4) which shows that Ala and Val residues exist in at least three different conformations. The presence of cross-peaks

with shifts in agreement with β-sheet structure supports the FTIR results that the CT domain must have undergone a structural transition from the helical soluble state to the fiber state with β-sheet content[17,54]. The presence of multiple secondary structures in the fiber state has been previously described for natural spider silk[43,44,55].

**Intermolecular interactions are formed within the CT fiber**

Silk fibers typically contain ordered β-sheet structures in which several protein chains are connected[23,43,44,52,56,57]. To detect intermolecular interactions, we prepared a second NMR sample in which recombinant CT domain with either $^{13}$C or $^{15}$N labeling was combined (50/50 ratio) in

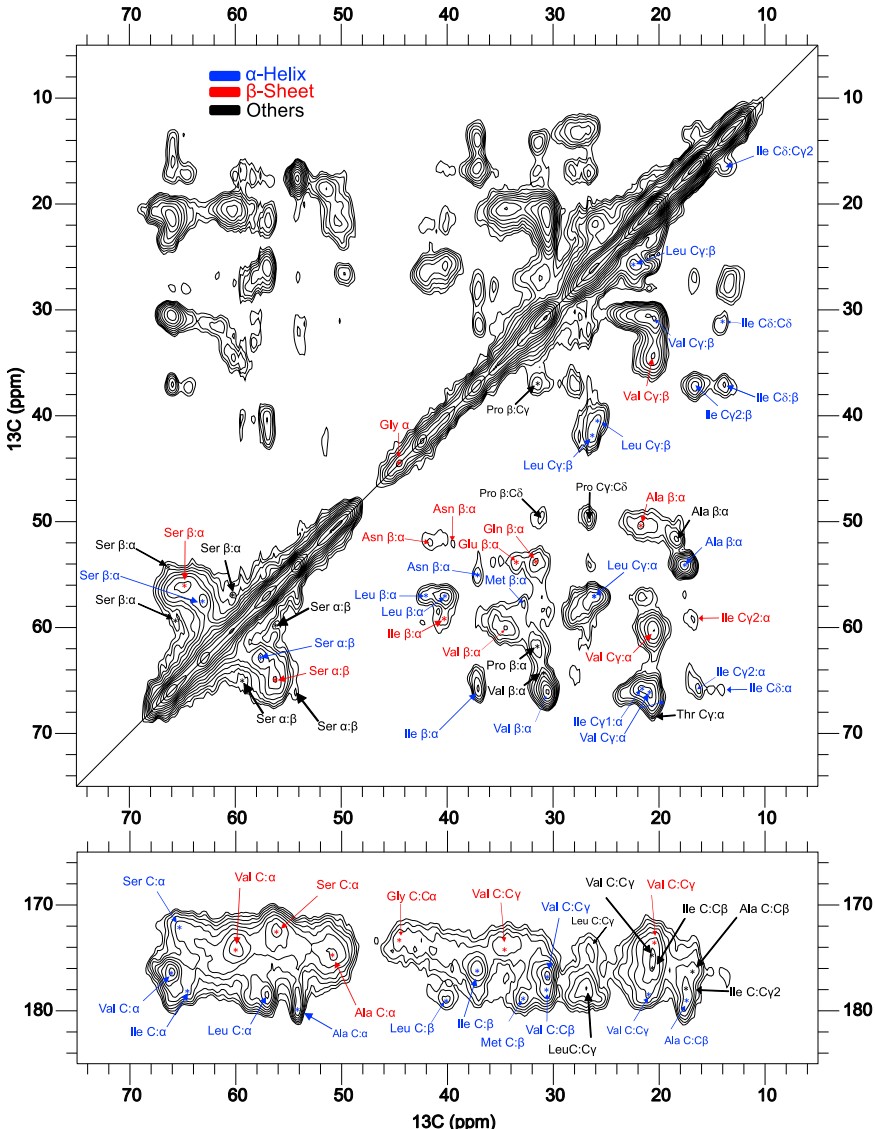

**Fig. 4 | 2D DARR spectra (50 ms mixing time) of CT fiber.** The secondary structure components of the amino acid residues of CT in fiber form based on the chemical shifts are indicated by the colors: β-sheet in red, α-helix in blue, and random coil in black.

solution before fiber formation. The sample was examined using proton-assisted insensitive nuclei cross-polarization (PAIN) experiments[58], in which the spectra show $^{15}$N-$^{13}$C contacts. Due to the suboptimal resolution in the $^{15}$N dimension, those 2D spectra were not useful. However, 1D $^{13}$C traces of the experiment show the peaks corresponding to the $^{13}$C nuclei involved in intermolecular contacts to $^{15}$N nuclei (Fig. S13b). The signals originate from $^{15}$N-$^{13}$C magnetization transfer, as the control experiment with zero power on the $^{15}$N channel did not give any signal. Both C′ and Cα signals are detected in agreement with close backbone contacts, possibly forming an inter-chain hydrogen bonded backbone network. Peaks originating from the residue types with strong signals corresponding to β-sheet conformation, i.e., Ala, Val, and Ser, are found in the PAIN spectra. Hence, the NMR analysis of the mixed sample shows that intermolecular contacts are formed in the fiber and that those contacts may involve β-sheet regions, as described for other types of silk[23,43,55].

### The structural properties of CT domain fibers indicate a key role of helix N°4 in the conversion from α-helix to β-sheet
The data presented so far clearly show that CT undergoes a structural conversion from α-helix to β-sheet upon fiber formation, although we

cannot say if the conversion involves parts of every CT molecule or the whole CT for parts of the sample. As sequential assignment of the NMR-data could not be achieved, and residues with potential β-sheet conformation can be found throughout the sequence (Fig. 5), we cannot determine which parts of CT that are responsible for the change in secondary structure. Therefore, we computationally explored the CT domain sequence for the propensity to form β-sheet fibrils (Fig. 5). This was done using the ZipperDB server[59] and the TANGO algorithm[60]. ZipperDB uses a database with known fibril-like protein structures to evaluate the propensity of a sequence to form steric zippers, which means two complementary β-sheets of the same segment that form the scaffold of amyloid-like fibrils. The algorithm is able to identify the main segments that could drive the formation of fibrils and act as hotspots for intermolecular clusters. The histogram bar in Fig. 5 shows the computed Rosetta energy for a modeled hexapeptide steric zipper for each residue in the CT sequence, with red/yellow indicating high fibril stability (a Rosetta energy below or equal to −23 kcal/mol). Since prolines are well-known secondary structure breakers, they are expected to be found at intersections or edges of β-strands[61]. The PDB coordinates for selected segment sequences were analyzed and the distances between the sheets of consecutive

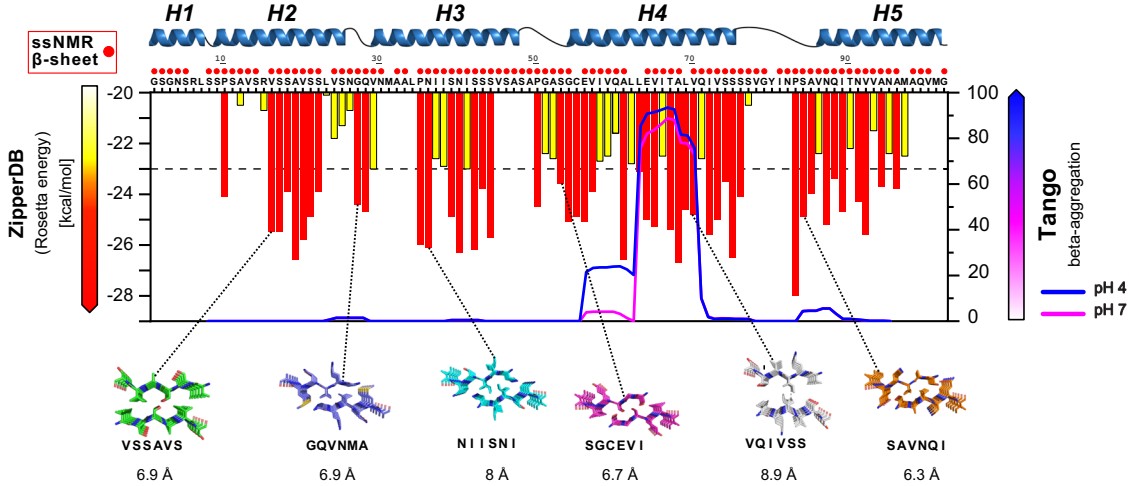

**Fig. 5 | Predictions of fibrillation-prone regions of the CT sequence.** The CT sequence with the positions of the helices in the model of the soluble CT shown as comparison. Amino acid residues with β-sheet chemical shifts from ssNMR are indicated in red. Computational analysis using ZipperDB fibrillation propensity and theoretical inter-sheet distances in combination with TANGO cross-β aggregation at pH 7 and pH 4 suggest hotspots for possible formation of β-sheets.

hexapeptides were measured. The results show that zippers with inter-sheet distances close to 9–10 Å, as observed by WAXS of CT fibers (Fig. 3a, c), are primarily found in the sequence of helix N°4.

The TANGO[60] algorithm inspects the protein sequence in fully denatured and solvent exposed conditions and predicts the cross-β aggregation propensity. Here, the highest score was found for the region within helix N°4. Taken together, the computational analysis combined with the experimental NMR and X-ray data suggest that helix N°4 have a higher propensity of forming β-sheet structure than the rest of the sequence (Fig. 5). This is intriguing since these helices are packed in a parallel manner (as coiled-coil homodimers), as previously mentioned (Fig. 1b).

### A synthetic peptide corresponding to helix N°4 (51-80) of the CT domain shows pH-dependent α to β structural conversion

To further investigate the role of helix N°4 of the CT domain, a peptide corresponding to amino acid residues 51–80, hereafter called helix N°4 peptide, were synthesized and analyzed. Matrix assisted laser desorption ionization -time of flight (MALDI-TOF) analysis of this peptide at pH 2 revealed the presence of a dimer, indicating disulfide bridge formation (Fig. S14). The CD spectrum of the helix N°4 peptide at pH 8–9 displayed two minima at 207 and 220 nm, thus a CD signature typically from α-helix[62]. However, reducing the pH to 6.1 resulted in a minimum at 218 nm, while more acidic pH conditions of 4.4 and 2.1 gave a minimum centered at 219 nm, canonical of β-sheet (Fig. 6a). Interestingly, the sample of helix N°4 peptide at pH 8 also shifted minima toward a single peak around 220 nm (β-sheet pattern) when kept for extended incubation time (16–20 h, at room temperature) (Fig. S15a).

Further, fluorescence microscopy analysis of the helix N°4 peptide with Amytracker suggests the presence of amyloid-like β-sheet structure (Fig. S9). Unlike α-helical structure, β-sheets require that several strands (from different parts of a peptide or from different peptide chains) associate to form the sheet. Hence, assembly of the peptide into supramolecular structures is likely associated with the structural transition. To investigate what types of morphologies that are associated with the β-sheet state of the peptide, we investigated the samples incubated at low pH (pH 3.1 and 2) by atomic force microscopy (AFM). The experiments revealed the presence of fibrillar structures with a height of a few nanometers, lengths in the micrometer scale, unbranched and periodic turns (Figs. 6b, S15). Taking also the β-sheet structure and the Amytracker results into account, these features indicate the formation of amyloid-like fibrils. Hence, the helix N°4 peptide is not only prone to undergo a α-to-β transition, it can also assemble into protein nanofibrils, contributing to the fundamental architecture of the crystalline parts of spider silk.

## Discussion

This study aims to investigate the role of the CT domain in the assembly of spider silk fibers using various techniques that allow studies of solid fiber samples. Previous ssNMR studies by Holland et al. have described the secondary structure of dragline silk, especially the environments of Gly and Ala that form unstructured 3₁₀-helices and ordered β-sheets, respectively[56]. Due to the significant contribution of Ala to β-nanocrystalline structures, the chemical shift of Ala Cβ is often used as a probe of the secondary structure in these regions. For instance, Creager et al. investigated the amino acid composition, hydrated backbone mobility, and variability of secondary structures using natural abundance ¹³C of collected dragline silk from five different spiders, assigning the Ala Cβ chemical shifts with predominant β-sheet fold, but also with a minor α-helix/coil contribution[52]. Our ssNMR data confirm the presence of both β-sheets and α-helices in the CT domain fibers. Notably, Gln, Glu, and Gly were found to form only β-sheet structures. We also show that intermolecular backbone contacts are present in the fiber. These findings are supported by FTIR data, which also suggests a predominant parallel orientation of β-strand with some anti-parallel contribution.

Previous studies combining ssNMR and X-ray diffraction of dragline silk have associated the protein sequence motifs with particular folding patterns and revealed distinct secondary chemical shifts assignments i.e., estimates of β-sheet content from specific amino acids, orientation of the Ala strands along the fiber axis, as well as size and morphology of the β-nanocrystalline structures[43,44]. The X-ray diffraction of the CT fiber presented herein shows a nanocrystalline component distinct from the β-poly(L-Ala) crystalline pattern while both distances are observed in the 4RepCT fibers. The main difference between these nanocrystalline components is the inter-sheet spacing, which is wider for the CT domain. The inter-sheet distances are associated with the packing of amino acid side chains. Hence, the X-ray data indicates that β-sheets formed by the CT domain involve amino acid residues with sidechains significantly longer than Ala. Such residues are indeed found among those in β-sheet configuration according to the ssNMR data. Moreover, hydrophobic amino acids such as Ala, Val, and Ile, which may play an important role in forming β-sheets, were

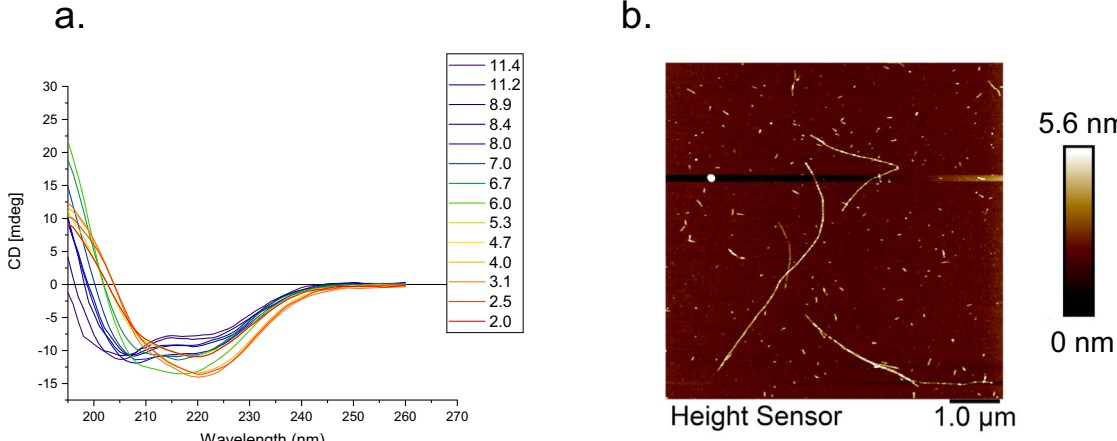

**Fig. 6 | The secondary structure evaluation of CT domain helix N°4 suggests a structural shift of the peptide to β-sheet. a** CD spectra of the helix N°4 peptide at different pH values. **b** Representative AFM image of nanofibrils formed by the helix N°4 peptide when incubated at pH 3.1.

found to be involved in intermolecular interactions disclosed by the PAIN experiment.

A number of findings from the integrated experimental and computational analysis points to the segment corresponding to the helix N°4 as pivotal for the structural transition into β-sheets (crystalline domain): (i) Glu were only detected in β-sheet conformation in the NMR assignment, and is only found in helix N°4; (ii) ZipperDB predicts high propensity of amyloid-like steric zippers in this region with several hexapeptides having predicted inter-sheet distances close to what is observed for CT fibers by WAXS; (iii) TANGO predicts helix N°4 to have the highest cross-β propensity in the whole sequence with a further increase at lower pH; (iv) the peculiar arrangement of helix N°4 for the two chains in the native dimer, including the disulfide linkage, position these fragments in a favorable conformation to start forming intermolecular β-sheets; and (v) two salt bridges stabilize the helical state at neutral pH but are broken upon pH shift, which would expose this segment to other spidroin molecules in the surrounding. To verify this assumption, a synthetic peptide corresponding to helix N°4, was prepared and analyzed. CD analysis corroborate the ability of helix N°4 to undergo pH dependent structural changes from α-helix to β-sheet. In addition, the nanofibrillar morphology and the binding of a dye developed for amyloid detection support the formation of ordered β-sheet structure with similarities to amyloid fibrils and silk nanocrystallites. The fact that the disulfide bridge seems to be intact in the CT fibers (Fig. S4) indicates that the Cys residues are positioned at the edge of the formed β-sheets. It has previously been shown that disulfide linkages within a parallel β-sheet would destabilize the structure, while if it is placed at the edge, it could even stabilize the conformation[63]. Hence, the disulfide may function as a "gatekeeper" in order to control the structural transition.

The spider silk fiber formation (in vivo and in vitro) relies on forces that will drive the silk crystalline/amorphous phases into an organized hierarchy. The changes in pH potentially play multiple roles for the molecular structure of spidroins. Acidification of the silk dope is a mechanism found in spider major ampullate silk glands, with a pH shift from 7.2 to 5.7[19]. Similar acidification of silk proteins in solution is an event that assists the fiber self-assembly process, speeds up the fiber nucleation-elongation kinetics and increases the β-sheet content[19,64–66]. Particularly, it is known that lower pH conditions play a role for the terminal domains: in the dimerization of NT[8] and destabilization of CT[22], mostly by effects on ionic interactions and polar groups[17,67]. Previously, pH has been described as an associated factor for fibrillation of the CT domain of diverse spidroins[65,68].

Likewise, the short silk segment LLEVVSA (partially homologous to helix N°4 peptide of current study), has been reported to form fibrils in an acidic pH range using 50% CH₃CN/0.05% aqueous trifluoroacetic acid[69]. Similar pattern of transition using CD spectroscopy has been reported when natural MaSp from *Nephila edulis* was heated from 20 °C to 55 °C[70]. In addition, acidic unfolding of the CT domain has been shown to lead to formation of amyloid-like fibrils reactive for the β-sheet binding dye thioflavin-T[18].

In summary, using solid-state methods, the analysis of recombinant CT domain silk fibers denotes a significant secondary structure transition, from mostly α-helix to β-sheet. WAXS investigations of the fibers confirmed that the wider inter-sheet distances associated with the CT domain are also found in 4RepCT, which support the extrapolation of the results also to more complex spidroins. Furthermore, the combination of experimental and computational methods assisted in identifying a specific segment within the CT domain (helix N°4) as a hotspot for fibrillation. A peptide corresponding to this hotspot region spontaneously shifts the secondary structure from α-helices to β-sheets and assembles into nanofibrils without the need for harsh chemicals or temperature increase, and the effect is enhanced by buffer acidification. On top of being necessary for breaking the salt bridges in the CT domain of spidroins, we show that concomitant acidification induces the formation of β-sheets in helix N°4. As the region is highly conserved among spidroins, the mechanism of secondary structure shift presented here (Fig. 7) could potentially be general for diverse types of spider silk and possibly also fibroins from other arthropods. We gained a deeper understanding of the importance of the CT domain structure for formation of β-sheets and crystalline regions which likely contributes to the reinforcing mechanics of silk. Also, it highlights the importance of the precursor helix N°4 transition; it represents one of the primary build blocks in the silk assembly hierarchy.

## Methods
### Expression and purification of recombinant MaSp1 proteins
Expression vectors were constructed to produce CT domain and 4RepCT, then they were transformed in *E. coli* BL21(DE3) cells. For 4RepCT and unlabeled CT, the cells were grown in Luria-Bertani medium containing kanamycin. For labeled CT, the cells were grown with minimal media supplemented with kanamycin, ammonium-¹⁵N chloride, and D-glucose-¹³C₆. The cultures were left to grow at 30 °C until the OD₆₀₀ reached 1.0. Subsequently, 0.5 mM isopropyl-β-D-thiogalactopyranoside was added to induce the protein expression and left 4 h at room temperature for incubation. The cell harvesting,

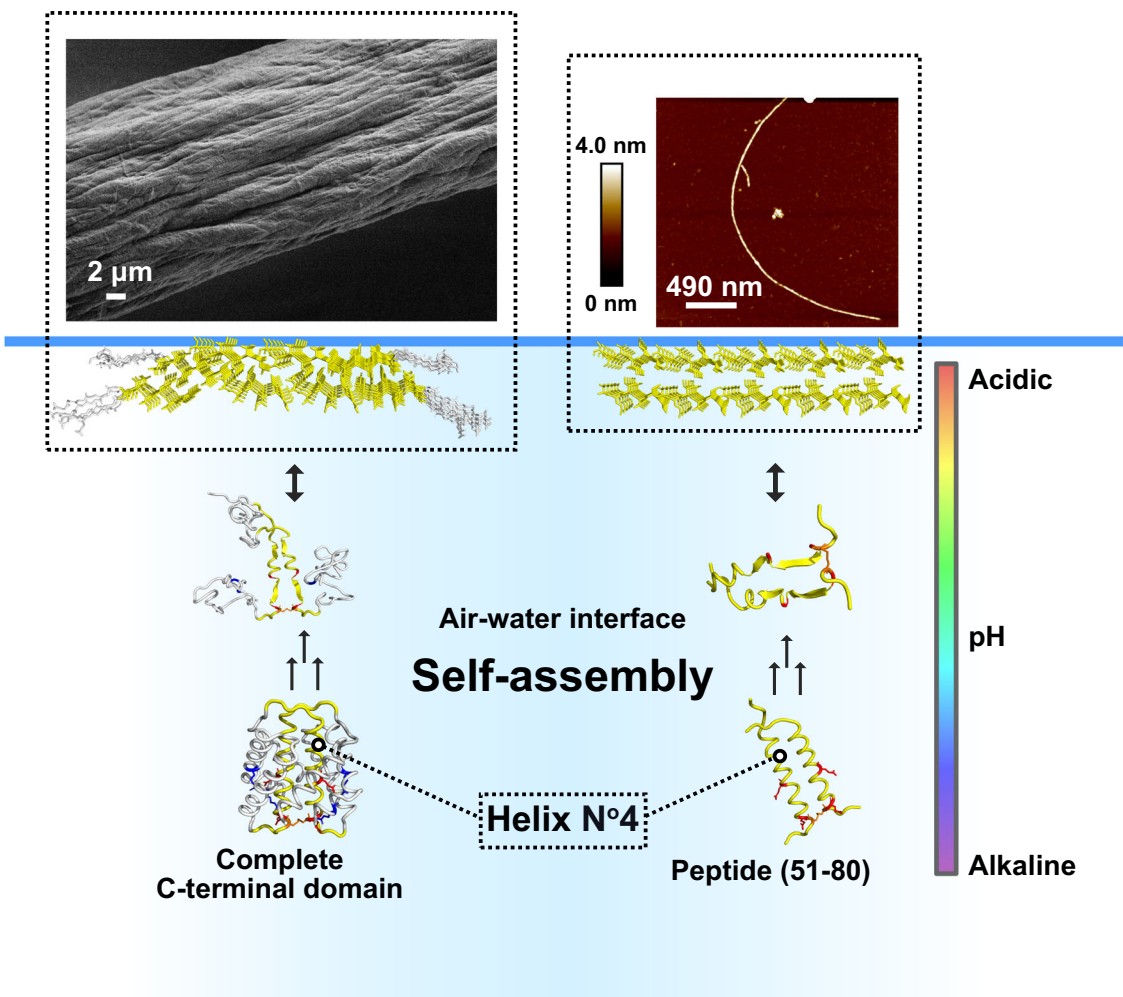

**Fig. 7 | The proposed model for α-helix to β-sheet structural conversion of the CT domain and the helix N°4 peptide.** The complete CT domain forms fibers, as presented with scanning electron microscopy (left side). The C-terminal segment helix N°4 refolds from α-helix to β-sheet to form a supramolecular β-sheet structure at a hydrophobic/hydrophilic interface. Helix N°4 (peptide CT$_{51-80}$) has been shown to form fibrils under acidic pH, as shown by AFM (right side). Glu is shown in red, Arg in blue, Cys in orange, and helix N°4 in yellow.

lysis, purification using native immobilized metal ion affinity chromatography, and proteolytic removal of the His$_6$-thioredoxin were performed[6].

### SDS-PAGE analysis

For CT in soluble form, the Laemmli SDS sample buffer (Bio-Rad), with or without the reducing agent dithiothreitol (DTT, final concentration of 10 mM), was combined with the sample and left for incubation at 95 °C for 10 min before protein electrophoresis. SDS-PAGE was carried out using Mini-PROTEAN® TGX™ precast gels 4–20% (Bio-Rad), 1x tris-glycine-SDS running buffer for 36 min, at 180 volts and 4 °C. As reference, the PageRuler (Thermo Scientific PageRuler™ Plus Prestained Protein Ladder, 10 to 250 kDa) was used. SimplyBlue SafeStain (Thermo Scientific) was used to stain the gels.

For analysis of the CT domain in fiber form, the fiber was first dissolved with 6 M urea. Laemmli SDS sample buffer (Bio-Rad) without reducing agent was added and left for incubation at 95 °C for 10 min before protein electrophoresis using Mini-PROTEAN® TGX™ precast gels 4–20% (Bio-Rad), 1x tris-glycine-SDS running buffer for 36 min, at 180 volts. Here, Bio-Rad Precision Plus Protein™ All Blue Prestained Protein Standard was used as ladder. Bio-Safe™ Coomassie Stain was used to stain the gel.

### Fiber formation

The stretch-relaxation cycles method was used for fiber formation[5,6,34,71]. In brief, the recombinant silk protein accumulates at the liquid-air interface, where shear forces and motions induce silk self-assembly at room temperature. The CT protein solution (3 ml à 1 mg ml$^{-1}$ in a 15 ml plastic tube) was placed on the rocking platform Duomax 1030–Heidolph Instruments set with an oscillation manner (5° tilt angle and 2 rpm) that facilitates the liquid-air interface interaction, being favorable for the formation of silk fiber against the tube walls.

### Attenuated total reflection Fourier-transform infrared spectroscopy

Attenuated total reflection (ATR) FTIR spectroscopy was conducted on the CT fibers using a Vertex 70 FTIR Spectrometer (Bruker) equipped with an HgCdTe detector and continuously purged with dry air to minimize water vapor interference. The samples were positioned on

the diamond crystal of a Bruker Platinum ATR unit and pressed onto the crystal with a piston in case of the fiber sample. The total spectral interval analyzed was between 3900 and 900 cm⁻¹, the spectral resolution was 4 cm⁻¹, and the zero-filling factor 2.

For the analysis, the amide I region (1700–1600 cm⁻¹) was selected, residual water vapor signals (when necessary) and a baseline were subtracted (straight lines with baseline points at 1800, 1740, 1720, and 1580 cm⁻¹). IR spectra were collected and pre-processed using the OPUS software (Bruker). Further evaluation was performed with the program Kinetics, kindly provided by E. Goormaghtigh (Université Libre de Bruxelles, Belgium).

For secondary structure analysis, component bands were simultaneously fitted to the absorbance spectrum and its secondary derivative in the 1700–1590 cm⁻¹ range[72]. The second derivative spectra were calculated using a Savitzky-Golay window of 21 points and were weighted with a factor of 300 in the fit. Initially, five bands with mixed Gaussian/Lorentzian band shapes were fitted in the 1710–1590 cm⁻¹ range. However, the optimal fit was obtained using three bands in the low wavenumber β-sheet region, i.e., in total seven bands in the 1710–1590 cm⁻¹ range (no baseline was fitted). The secondary structure content was estimated from the relative band areas assigned to the different secondary structures. The data was plotted using the software OriginPro[73].

The spectral ranges used for the assignment of secondary structures[37–39] were: β-sheets 1610–1641 cm⁻¹, anti-parallel β-sheets 1700–1690 cm⁻¹, α-helixes 1648–1657 cm⁻¹, and other structures 1642–1686 cm⁻¹ (i.e., all *other* secondary structures: irregular, turns, bends, other helix types). Secondary structures other than α-helixes or β-sheets are expected to produce broad bands, which are relatively weak in second derivative spectra. Therefore the 1650 cm⁻¹ band was assigned to α-helixes.

In order to discriminate between parallel β-sheets and anti-parallel β-sheets, we used the β-sheet organizational index, described by ref. 40. To calculate the β-sheet organizational index, the spectra were subjected to a slightly different protocol for Fourier self-deconvolution using a Lorentzian deconvolution factor with a full width at half height (FWHH) of 20 cm⁻¹ and a Gaussian apodization factor with FWHH of 13.33 cm⁻¹ to obtain a resolution enhancement factor $K = 1.5$. After a baseline subtraction using baseline points at 1710 cm⁻¹ and 1580 cm⁻¹, the deconvoluted spectra were fitted with 8 component bands according to the original publication. The ratio between the maximum absorbance of the high and low wavenumber β-sheet bands (after fitting) is considered the 'β-sheet organizational index'. This study uses the 1695 cm⁻¹ band as high wavenumber band and the 1624 cm⁻¹ band as low wavenumber band.

## X-ray fiber diffraction

Synchrotron radiation (SR) experiments were performed at the ESRF-ID13 beamline using a monochromatic beam of $\lambda = 0.09996$ nm[74]. The beam was focused by crossed mirrors to an $0.7 \times 0.7$ (μm)² spot at the sample position with a flux of $\sim 5 \cdot 10^{10}$ photons/s. A Frelon CCD detector[75] with 100 mm converter screen, $2\,K \times 2\,K$ pixels of $51 \times 51$ (μm)² size, and 16 bit readout was used. The detector was $2 \times 2$ times binned. A short section was cut by microscissors from a piece of fiber and attached by fast glue to a tapered glass capillary, with the fiber vertically oriented. The capillary was fixed by a magnetic support piece to the beamline scanning goniometer. The fiber section was kept during diffraction experiments at 100 K by an *Oxford Cryosystems* nitrogen cryoflow system. No icing was observed during the experimental period. The sample-to-detector distance was determined as 76.3 mm using an Al₂O₃ calibration sample. In order to improve the counting statistics, the diffraction patterns were averaged over several patterns measured during a horizontal scan across the fiber. The corresponding averaged background patterns were subtracted. Data collection was done by 2D raster-scans using 5 or 10 μm steps. After

every raster-step a WAXS-pattern was collected. A typical collection time for a diffraction pattern was 1 s. The scattering measured during a raster-scan outside of the fiber was subtracted as background. The FIT2D program was used for data analysis and data display[76].

The crystallinity index ($X_c$) is approximated as:

$$Xc = \frac{\sum I}{(\sum I + \sum B)} x100 \qquad (1)$$

Where is $\sum I$ the sum of Bragg peak intensities and $\sum B$ the diffuse background scattering.

For the SR experiment, a slightly different CT construct was used, see supporting information Figs. S2, S16. Therefore, complementary X-ray experiments were carried out for fibers from the current construct, using a Cu Kα X-ray source, $\lambda = 1.5418$ Å. An Eiger R 1 M Horizontal detector was used and the sample-to-detector distance was 110.8 mm. Experiments were conducted with an acquisition time of 10 min with 3 frames for 4RepCT and 18 frames for the CT domain. The obtained diffraction patterns confirmed that no major differences can be observed between the two CT constructs (Fig. S16).

## Solid-state nuclear magnetic resonance experiments (ssNMR)

Dry fiber samples for ¹³C natural abundance ssNMR experiments were packed into 4.0 mm zirconia MAS rotors and a few microliters of water was added to hydrate the sample. Kel-f inserts (vol. 30 μL) with sealing plug and screws were used to keep sample wet during the experiments. Solid-state NMR experiments were performed on samples of CT fibers (sample I) and 4RepCT fibers (sample II) using a Bruker Avance III HD 500 MHz NMR spectrometer equipped with a 4.0 mm MAS probe (Bruker Biospin). The ¹³C CP-MAS NMR experiments were performed at a spinning rate of 9 kHz with a sample temperature of 298 K, ¹H 90° pulse length of 3.3 μs, a 2.0 ms contact time, a recycle delay of 2.5 s, and an applied Spinal-64 decoupling[77] on ¹H at 75.8 kHz.

The isotopically labeled fibers were packed into 3.2 mm zirconia MAS rotors using a spiNpack ultracentrifugal packing device (Giotto Biotech, Italy), which was then placed in a Beckman Coulter Optima L-100 XP with swinging-bucket SW 28 Ti ultracentrifuge rotor and the sample was first centrifuged at $141,000 \times g$ over 1 h. About 1.5 mL of the buffer was used to rinse the original sample tube and centrifuged again before the rotor was sealed for MAS NMR measurements. To restrict the samples to the central third of the rotor, a thick-bottom 3.2 mm rotor with a 12 μL Teflon spacer above the sample was used. Uniformly ¹³C-¹⁵N-labeled CT domain protein fiber (sample III), and a mixture of ¹³C-labeled CT and ¹⁵N-labeled CT protein fiber 1:1 (sample IV) were prepared. Experiments were performed on a 850 MHz Avance III HD spectrometer (Bruker Biospin) equipped with a 3.2 mm triple resonance MAS probe. All experiments involving labeled fibers were performed at a MAS spinning rate of 17.5 kHz with a sample temperature of 278 K unless stated otherwise. To gain access to intermolecular and intramolecular structural information of the fibers, a series of ¹³C-¹³C 2D DARR (Dipolar Assisted Rotational Resonance)[78] NMR data were acquired with 25, 50, 100, 200, and 400 ms mixing times. The following spectral parameters were used to setup the experiments: ¹H 90° pulse length of 3.0 μs, ¹³C 90° pulse length of 3.5 μs, ¹H-¹³C CP was implemented using a linearly ramped RF amplitude centered around 65 kHz on the ¹³C channel while a constant RF field of 84 kHz on the ¹H channel for 1.5 ms (−1 spinning side band), a recycle delay (RD) of 2 s and an applied Spinal-64 decoupling on ¹H at 84 kHz. DARR spectra were recorded by acquiring 192 points in the t1 dimension with dwell time of 11.0 us and 40 scans per free induction decay (FID).

For ¹⁵N assignments, both one-bond NCa and NCO, and multibond NCX correlation data to side chain carbons were acquired. The double-CP (DCP)[79] experiment correlates ¹³C and ¹⁵N chemical shifts. During the initial CP transfer (¹H to ¹⁵N) in NCO and NCA double CP experiments, a constant ¹H RF amplitude of 59 kHz is applied along with a

linear ramp from 41 and 46 kHz for $^{15}$N RF field over 0.4 ms (contact time). In the second CP step, the magnetization is transferred from $^{15}$N to $^{13}$C selectively by setting the carrier frequency to be on-resonance with either CO or Cα. For the NC transfer, from $^{15}$N to $^{13}$C, a constant RF amplitude of 44 kHz and 26.5 kHz are applied on $^{15}$N and $^{13}$C (-1 side-band), respectively, along with $^1$H CW decoupling at 70 kHz for 2.5 ms. A Spinal-64 decoupling on $^1$H at 83 kHz was applied during $^{13}$C acquisition of 24 points in the t1 dimension with a dwell time of 11.0 μs and 800 scans per FID. To establish long distance intermolecular contacts between $^{13}$C and $^{15}$N, proton assisted insensitive nuclei (PAIN)[58] CP experiments were performed on sample IV. PAIN-CP spectra were measured with a 4 ms mixing time, using RF fields of 45 and 44 kHz for $^{13}$C and $^{15}$N (2.5x spin rate 17.5 kHz), respectively, and $^1$H RF field at about 55 kHz.

$^{13}$C NMR chemical shifts were calibrated externally to tetramethylsilane, TMS ($\delta_{iso} = 0$ ppm) by adjusting the field to align the high frequency peak of adamantane at 38.48 ppm. This setting establishes the $^{15}$N chemical shift of powdered NH$_4$Cl at 39.3 ppm relative to liq. NH$_3$. The reported chemical shifts were corrected for the difference between 2,2-dimethyl-2-silapentane-5-sulfonic acid (DSS) and TMS by adding 2.67 ppm[80]. All data was processed using the TopSpin 3.5 software package. The 2D spectral analysis and peak assignments were performed with CCPNMR 2.5.2/3.2.0[81]. All reported shifts were validated in the full set of recorded spectra. Assignments with apparent inconsistencies were excluded from the final set of reported assignments. Interpretations of chemical shifts in terms of secondary structure were based on the "Wishart" protein secondary structure parameters[53].

### Solubilization, analysis, and subsequent fibrillation of the helix Nº4 peptide

The helix Nº4 peptide, CT$_{51-80}$ (Ac-GASGCEVIVQALLEVITALVQIVSSSSV GY-NH2) was purchased from AlexoTech AB (Umeå, Sweden). The peptide as lyophilized powder was reconstituted on the same day of the experiments, using 20 mM tris pH 12 (1 mg/ml), further individually aliquot, diluted and pH adjusted to suit the experiments. The peptide was analyzed with MALDI-TOF mass spectrometry, using the matrix α-Cyano-4-hydroxycinnamic acid and processed with Applied Biosystems - 4800 MALDI TOF/TOF™ Analyzer. CD spectroscopy analysis was conducted immediately after reconstitution in 20 mM tris acid (or neutral) solution and after incubation for up to 20 h at room temperature.

### CD spectroscopy of CT domain and the helix Nº4 peptide

Samples with concentrations of 59–65 μM (0.18–0.2 mg/ml) for the helix Nº4 peptide and 0.2 mg/ml for the full-length CT domain were used for the CD experiment. The peptides and CT domain were analyzed with a Chirascan CD spectrometer (Applied Photophysics) and a 1 mm quartz cuvette. The temperature was maintained at 20 °C. The wavelength scan range was 190–260 nm (1 nm steps), the bandwidth was 1 nm, and the time per point was 1 s, with triplicate measurements for each sample. Data was smoothed using Savitzky-Golay filter with 7 points of window. The deconvolution analysis of CT-domain in solution was performed using the BeStSel[82] software.

### Computational sequence analysis, homology model and fibrillation hotspots identification

The input sequences can be found in Fig. 1 and supplementary material (Fig. S2). For the CT domain of MaSp1 sequence conservation analysis, Consurf [83] with standard configuration was used: the Bayesian method of calculation and HMMER method for sequence identification and alignment with minimal 35% identity for homologs. The pipeline begins with the query sequence BLASTed[84] against UNIREF-90 database[85], presenting 217 hits. Then the redundancies were removed and multiple sequence alignment (MSA) performed on 143 unique sequences in order to obtain the phylogenetic tree. Based on the tree and MSA, the algorithm computes the position-specific evolutionary rates using an empirical Bayesian approach, assigning and ranking into nine conservation degrees. The analysis was performed using the server https://consurf.tau.ac.il/.

The algorithm from SWISS-MODEL[86] was used to obtain CT domain structure homology model, with a sequence identity of 60.4% and the main template PDB: 2KHM, https://swissmodel.expasy.org/. To identify fibrillation prone segments the servers of ZipperDB[59] and TANGO[60] were used: https://services.mbi.ucla.edu/zipperdb/ and http://tango.crg.es/, respectively. For TANGO analysis, the standard configuration was: 298.15 K, ionic strength = 0.02 M, concentration = 1 M, for both pH 4 and pH 7. The normalized Kyte & Doolittle hydropathicity was determined with https://web.expasy.org/protscale/.

### Atomic force microscopy of silk the helix Nº4 peptide

The helix Nº4 peptide fibrils were imaged by means of AFM (Dimension FastScan microscope, Bruker) using Nanoscope Software (version 9.1, Bruker) and FastScan A probes (Bruker) operating under tapping mode. Samples were prepared by diluting a stock solution of fibrils (0.1 mg/ml, pH 3.1 and 2) between 10 and 1000 times in 1 mM HCl. 25 μL of the diluted samples were pipetted onto freshly cleaved mica and incubated for 10 min at room temperature. After incubation, aliquot excess was rinsed with filter-sterilized milli-Q water and dried using a stream of compressed air. Flattening of the height topology data and removing of the tilt and scanner bow were achieved using Nanoscope analysis software (Version 1.9, Bruker).

### Fluorescence microscopy and β-sheet structure detection

The fluorescent dye Amytracker 630 or 680 (Ebba Biotech AB) (diluted 1:1000) was added to the peptide helix 4 (1 mg/mL) in solution or CT (3 mg/mL) after fiber formation. For the helix Nº4 peptide, 20 μL solution was placed on a microscope slide and then imaged using an inverted fluorescence microscope (Nikon Eclipse Ti, Nikon, Netherlands) and the filter "Texas Red" (Nikon). The CT fiber was moved to a 24-well plate and imaged while still in solution using a Leica DMI6000 B microscope.

### Scanning electron microscopy

The CT fibers were placed on top of conductive carbon tape and coated with 20 nm of gold through metal evaporation (ProvacPAK 600 Coating System, Germany). The samples were then imaged using SEM (Gemini Ultra 55, Zeiss, Germany).

### Reporting summary

Further information on research design is available in the Nature Portfolio Reporting Summary linked to this article.

## Data availability

Source data files are available at https://doi.org/10.5281/zenodo.11110182 [https://zenodo.org/doi/10.5281/zenodo.11110182]. Source Data are also provided with this paper. The existing structure 2KHM was used in this study. Source data are provided with this paper.

## Code availability

All technical details for implementing the modulations are enclosed in the Methods section. Codes are available from the authors upon request.

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

## Acknowledgements

Financial support from the Swedish Research Council (grant number 2020-03329 to C.L.) is acknowledged. The FTIR spectrometer was funded by the Knut and Alice Wallenberg Foundation. We thank E. Goormaghtigh (Université Libre de Bruxelles, Belgium) for providing the program Kinetics, Sara Zillen for CD data collection of the C-terminal domain silk in solution, Wei Zhao and Anita Teleman from RISE Research Institutes of Sweden for the data collection of supplementary X-ray diffraction of silk fiber experiments. We acknowledge the efficient help of ID13 beamline staff Manfred Burghammer and Emanuela DiCola during the experiments.

## Author contributions

C.L. and M.H. conceived the study and coordinated the work. D.H.D.O. performed most of the experimental work and data analysis. V.G. and

T.S. performed NMR experiments. L.G. performed microscopy experiments. R.S.P. performed AFM experiments. C.R. performed X-ray experiments and assisted in data analysis. A.B. performed FTIR experiments and assisted in data analysis. D.H.D.O. wrote the manuscript under supervision by C.L. and M.H.

## Funding

## Competing interests
M.H. has shares in the research company Spiber Technologies AB. L.G. is employed by the research company Spiber Technologies AB. The remaining authors declare no competing interests.
