## [Peer Review File · Nature Communications]

Reviewers' Comments:

Reviewer #1:

Remarks to the Author:

The study in the manuscript "Untangling spider silk secrets: The structural basis of alpha-helix to beta-sheet conversion 2 of the spidroin C-terminal domain during fiber assembly" by De Oliveira et al focuses on unraveling the assembly mechanisms of spider silk, specifically the Major ampullate Spidroin 1 (MaSp1) protein's C-terminal (CT) domain, which is crucial for the self-assembly of silk fibers. The researchers discovered that the CT domain can independently form silk-like fibers without the repetitive domain, offering a unique opportunity to gain insights into this process as well as the process of natural spider silk fiber formation. Using techniques such as Fourier transform infrared spectroscopy, X-ray diffraction, and solid-state nuclear magnetic resonance, the study thoroughly examined structure of the macroscopic fibers. The results revealed a helical to β -sheet transition in the fibers. Computational algorithms identified the sequence corresponding to helix No4 as the likely initiator of this structural conversion.

Overall, the study is sound, and the manuscript well written providing high quality results. However, some issues in the study require clearance. I recommend the publication after minor corrections, as follows:

Introduction should provide more comprehensive overview on the topic. Herein, the introduction part (lines 47-74) is actually very specifically related to one type of NT, CT and Rep domains used in the research group, although many statements sound general i.e. valid for majority of the studied domains from different species. For example:

Lines 56-57. "The NT domain cannot form fibers, neither on its own nor followed by a part of the repetitive..." The literature provides examples where a repetitive domain with a NT domain could be spun into fibers from aqueous solutions. Hence described phenomena is rather related to the specific NT and Rep sequences. Authors should comment on this and extend the part providing more examples on to the issue of fiber formation from recombinant proteins containing NT and/or CTs from different spider species. The source (spider silk species) of the studied or compared NT and CT domains should be noticed in the introduction and whole manuscript for clearness.

Lines 86-91: Please make clear on what type of material the analyses have been made i.e. natural spider silk fibers or those made of recombinant spider silk proteins.

Results:

Line 101: the term "fiber format" in whole manuscript sounds confusing. I recommend using fiber(s) fiber (-like) morphology or fiber form.

FT-IR characterization:

In Figure 2 the Amid I band of the CT domain before transformation (in solution) and after the transformation (in the fibers) should be presented.

The deconvolution of the Amid I band is confusing. 2nd derivative shows presence of 4 possible peaks (according to the shifts possibly assigned to parallel beta-sheets, alfa-helices, turns and antiparallel beta-sheets (from left to the right)), hence it would be more reliable to fit those 4 peaks into the envelope the amid I and afterwards assign the structure according to data from literature (the literature source for the assignment is missing e.g., in Table S6). Currently, the appearance of the peaks in the Figure 2C (4 for beta-sheets, 2 for "other" and 1 for alfa-helices) is unclear. The correct deconvolution-based evaluation of the spectra is however suggested in Figure S3, hence the same should be used in the Figure 2C.

Figure 3, what kind of steric zipper structure are shown in Figure 4C and D, an arbitrary or particular ones?

The CT domain is apparently stable at different pH as the CD spectra in Figure S1 clearly show alfa-helical structure in the pH range 12.8-4.8, whereas the hydrophobic peptide from the Helix Nr. 4 readily transforms into beta-sheets, hence the relation of this experiments in Figure 6A to the explanation of the observed CT transformation is rather elusive, especially because the pH shift plays no role in the experiment presented to the CT and CT-Rep transformation into fiber. Please clear this. How stable is the CT domain and CT-Rep at pH range 7.5-4.5 (thermal stability at different pH)? Also, for the demonstration of the differences in the pH triggered transformation, Figure S1 should be included into Figure 6, whereas Figure 6B could be excluded in supporting information.

The beta-sheet rich structure in the CT fibers as well as those the peptide fibrils are proposed to be

amyloid like. Hence experiments typical for amyloids such as ThT binding onto fibrils or “green birefringence” of Congo red on the fibers should be demonstrated.

Reviewer #2:

Remarks to the Author:

De Oliveira present an interesting study that provides a valuable expansion to knowledge of spider silk fibre formation. Namely, the role of the nonrepetitive C-terminal domain (CT) in self-assembly is probed in major ampullate-silk based constructs, with focus on the CT itself from *E. australis*. A series of analyses of this CT are presented in the solid-state (ATR-FTIR, SRD, solid-state NMR) with underpinning by computational analyses (homology modeling, amyloid propensity prediction, evaluation of modeled peptide energy by Rosetta). In some instances, direct comparison is carried out with the partial spidroin construct 4RepCT, which is composed of four copies of poly-Ala/Gly stretches from the repetitive domain fused to the CT. The rationale for the study herein is that the CT has been shown to self-assemble previously into fibres (albeit with lower stability than 4RepCT), providing a means to evaluate the putative involvement of the CT in triggering of fibre formation through a role in alignment of the repetitive domains. The authors convincingly demonstrate that CT-based fibres have mixed α -helical and β -sheet composition and show that the β -sheet components are distinct from those of the Rep in terms of the observed inter-sheet spacing, consistent with residues having longer sidechains. Based upon chemical shift evaluation in solid-state NMR, the structural transition is linked to helix 4 of the CT, a segment of the CT that – based upon homology modeling – would be expected to form an interface with helix 4 of the second protomer in a dimer where the helix 4’s of each protomer are aligned in a parallel fashion (at least in instances where this is disulfide linked). There are definitely some sections which would benefit from a little bit more proofreading and polishing, but as a whole this is both a well-written and easy to follow manuscript. Notwithstanding my generally positive assessment of the work and impression that this is both of broad interest and valuable to the silk field more specifically, I do have some concerns that I feel should be addressed prior to consideration for publication.

First off – most importantly – with all of the focus on the CT dimer state and potential for the (putatively) disulfide linked helix 4 to influence self-assembly, there appears to be a major missing set of information with respect to evaluation of the state of the Cys residues in the full-length CT and synthetic peptide. The authors appear to be presuming there is a disulfide (e.g., Figure 7), but is there any experimental evidence to demonstrate whether the CT being used in a given sample/condition is in a disulfide linked dimeric state (or not)? I may have missed this, but I do not see any mention of this anywhere or experimental demonstration of this. Clearly, this has important ramifications for all of the interpretation. Related to this (but independent of demonstration of dimer formation and stability), for sample IV (mixed ^{13}C -labelled + ^{15}N -labelled protein), more detail would be beneficial with respect to (1) oxidation state and (2) when proteins were allowed to disulfide link (i.e., would mixed $^{13}\text{C}/^{15}\text{N}$ -labelled disulfide linked dimers be expected at statistical prevalence, or was dimer formation carried out prior to mixing of the proteins?) In general, some additional detail about the state of the protein in terms of the disulfide is needed in all sample preparations. Likewise, while the CT peptide is discussed in terms of disulfide formation, there is no mention of this peptide being prepared in the oxidized state and/or oxidized prior to usage to form disulfide-linked dimers, nor is it shown to be a dimer by any experimental methodology. On page 17 (point iv, lines 357-359), can further comment/discussion be made about the ability of the S-S linkage to be accommodated in the sheet (it’s noted that the disulfide provides a favourable situation for beta-sheet formation, but disulfides actually tend to perturb at least one strand in a sheet and are relatively rarely observed in beta-sheets – the study by Almeida et al. (2012, JACS, 134: 75) may be beneficial to consider as part of this expanded rationalization?) Finally, one related note is that Cys isn’t assigned in the solid-state NMR data – could the unassigned cross-peak at ~38 ppm / 58 ppm be Cys (assuming independent evidence of disulfide linking, before using this cross-peak as evidence of the cystine state) and if so what would the inferred secondary structure be? (I recognize this is only a single residue, so perhaps the signal-to-noise is not sufficient to pick this up.)

Over and above this key concern, some other issues that I would suggest in terms of required

revisions are:

1) It is a bit unclear how the ConSurf analysis described in the methods was used and what the source of the 143 unique sequences noted as being used for "calculation" (what sort of calculation?) were vs. the 217 "hits" that were noted. Was this used as part of the homology modeling with SwissModel, or is that strictly homology model? What kind of quality metrics/statistics are available for the homology model (over and above the reported % identity)? Note in methods (line 568): "determined" should be determined.

2) It would be great to have a more direct overlay of the 2nd derivative and absorbance data if Figure 2 (or at least some guide-lines to allow for comparative assessment of the components noted in each.)

3) Although an optical micrograph of the CT-derived fibres is presented in Figure 2 and an SEM micrograph is presented Figure 7, details of the fibre morphology and features (as well as of the microscopy methodology) are not provided. Are these truly "silk-like" fibres that recapitulate previous studies of the CT?

4) p. 9, line 177 – presumably this should be a reference to Figure 1 not Figure 4?

5) p. 9, para. 2 – is it possible to show a more direct comparison to Rep domain-based fibre diffraction (or, lacking this, wildtype fibre diffraction where the repetitive domain would presumably be dominating) in terms of backing up the statement that 4RepCT is a superposition [convolution may be better] of the CT domain fibre and the β -poly-(L-Ala) structure? From the data provided in Fig. 3 alone, this statement is hard to assess.

6) p. 10, lines 205-207 – it is a bit unclear why the aromatic content would be "lower" in 4RepCT than in a natural silk, at least if derived from the same protein. (i.e., per unit mass in the MAS rotor, the proportion of aromatics should be ~similar, shouldn't it?)

7) p. 10, para 2 – for arguments about relative abundances etc., it would be easier to follow and assess this if the authors present a table or histogram of amino acid #'s/proportions [could be supplementary info] in the Rep vs. CT domains.

8) p. 10, Lines 221-222 – the inference is made that there is disorder in the fibre state due to the fact that ^{15}N spectra were poorly resolved. This seems like a large leap of logic, given that there could be a variety of causes of broadening and loss of resolution over and above disorder.

9) p. 10, lines 224-226 and subsequent to this – there is no detail provided about how chemical shifts were evaluated with respect to secondary structure inferences, nor about the "robustness" of a given assignment. I am also concerned about the ^{13}C referencing noted, where TMS was noted to be the 0 ppm reference – biomolecular secondary structure evaluation is typically evaluated with respect to DSS at 0 ppm, which is in turn substantially offset from TMS for ^{13}C , so if the data presented are indeed referenced to TMS then this would lead to a consistent offset in chemical shifts that would bias/modify secondary structure inferences.

10) p. 10, lines 227-229 – what is the rationale that "additional residues" are observed through longer (or different) DARR mixing times? I would expect further spin diffusion with a longer mix (i.e., observing dipolar species that are further separated) but not necessarily the ability to see "additional" residues

11) p. 12, Figure 4 – fairly major conclusions are drawn from the Gln/Glu assignments that are made (e.g., p. 16, lines 332-333). However, these appear to be based on a single (relatively weak) cross-peak for each residue type. One might also argue that there's possibility that one of the CB shifts is actually a CG shift, for example, in that perhaps only one of these amino acid types is observed? This comes back to point # 10 about the method of assignment and robustness of assignments

12) p. 13, line 258 – would “undergoes a structural conversion” be better worded as “undergoes a partial structural conversion”? This appears to be the main conclusion?

13) p. 14, Figure 5 and related discussion – the inclusion of Rosetta energies seems out of place (until one realizes this is output from ZipperDB) – some additional clarification as to the methodology used and information obtained would be beneficial. It may also be helpful to have the color scale legend used for Rosetta energy scaled in such a way that the length of the arrow matches the numerical values on the y-axis to a given color or else to present this differently so that the coloured arrow is not directly aligned with the y-axis.

14) p. 15, lines 297-298 – the noted behaviour was not entirely clear on first read. I would suggest rewording this.

15) p. 16, para. 1 – there’s noted to be a pH-dependent trend with respect to a progressive beta-sheet transformation. Is there a [clear] pH-dependence of the kinetics of this transformation? (It’s not entirely clear what type of pH-dependent behaviour was observed.) It’s also not clear why the data at high pH are not shown?

16) p. 16, para. 2 – the AFM data certainly appear to show fibrillar species, but there is a concern that these micrographs show imaging issues (e.g., loss of contact) and have height scales that start from substantial negative values. It’s not clear in the methods that these experiments were repeated, nor is it clear that these are “representative” images as opposed to a single pair of images collected for a single fibre at two resolutions. All in all, as presented, the AFM data are not highly compelling.

17) p. 20, para. 1 – the expression and purification protocols are somewhat sparse. (What form of fusion protein was used? How was cleavage carried out? What level of purity was achieved [i.e., as demonstrated by SDS-PAGE, for example]?) Perhaps this is all exactly as described in ref. #6, but the level of detail is still not sufficient and the protein purity following cleavage is important to document and demonstrate.

18) p. 22, lines 488-490 – the statement that the two constructs have no major differences in diffraction pattern is a bit hard to assess in direct comparison of Figure S8 and Figure 3, where the annotated features are not always visible in Figure S8 and with the differences in presentation with respect to q values plotted and apparent features that are visible in Figure S8(c) that differ from those in Figure 3.

19) pp.22-23 – it comes as a bit of a surprise in the methods that the fibres studied by solid-state NMR are “hydrated” (p. 23). This should be more clearly noted in the results/discussion in terms of clarification of the manner in which the solid-state NMR samples are prepared.

20) Para. spanning pp. 23-24 – there’s a lot of detail provided on backbone assignments when it seems this process wasn’t actually carried out. This could perhaps be shortened or removed as it wasn't applied?

Reviewer #3:

Remarks to the Author:

The work reports that C-terminal domain in major ampullate spidroin 1 is capable to form fibrillar structures without the presence of MaS1 repetitive domains.

Overall, this is a highly rigorous and useful paper that uses an ensemble of spectroscopy methods, including FTIR, CD, NMR, X-ray for analysis of CT assembly.

I believe there are extra minor details that could be added to improve the clarity of the paper and the certainty of the interpretations.

1) Authors have modeled overall folding of CT domain and confirmed this experimentally with circular dichroism analysis (Figure S1). However, Fig. S1 depicts only original spectra with no data

interpretation. I suggest to add chart showing the fraction of beta-sheet, random coil/helical content based on depicted CD analysis. Also, how authors explain that CD spectra of CT at pH 4.8 and pH 12.8 almost perfectly overlap (at least for my unequipped eye)? How relevant those extreme pH values to actual pH gradient in spider silk gland?

2) If I understood correctly the fiber has been formed via shaking (stretch-relaxation cycles method)? If not please expand experimental details. If yes, can authors provide shaking conditions in the materials and methods section.

3) FTIR fitting... Can authors provide ranges for beta-sheet, helices peaks assignment. Also, how "other" secondary structure has been interpreted. Has it been assigned to helical conformation or to beta-sheet? "Other" conformation constitute 15.55% of the overall structure and can affect data interpretation. The sample of FTIR (ATR) analysis contained only fibrillated peptide or mixture of fibrillated and monomeric peptide? There are some useful papers describing FTIR analysis of the silks that I recommend to cite if not cited yet (1) Nat Commun 13, 7856 (2022).

<https://doi.org/10.1038/s41467-022-35505-w>; 2) <https://doi.org/10.1002/app.21346>; 3)

<https://doi.org/10.1016/j.vibspec.2007.10.003>)

4) Experimental section on CD data interpretation/fitting/analysis is missing.

5) Again how chosen pH values for peptide fibrillation are relevant to physiological conditions of spider? Author should comment on this in the main text.

6) Discussion section on FTIR data. Can author comment on the relative ratio between parallel and anti-parallel beta-sheet content.

7) X-ray data: Out of curiosity I am wondering whether authors succeeded to detect the pattern of H-bond network? (see relevant work: Commun Chem 4, 62 (2021).

<https://doi.org/10.1038/s42004-021-00494-2>) What was the solvent for the peptide solubilization? How pH has been adjusted?

KTH – Stockholm March 2024

First of all, we would like to thank all the reviewers for their valuable feedback, which greatly helped us to improve the manuscript. We are also very grateful for the encouraging and positive comments from all the reviewers. Here follows a point-by-point response to all issues raised by the reviewers.

Reviewer #1

The study in the manuscript “Untangling spider silk secrets: The structural basis of alpha-helix to beta-sheet conversion 2 of the spidroin C-terminal domain during fiber assembly” by De Oliveira et al focuses on unraveling the assembly mechanisms of spider silk, specifically the Major ampullate Spidroin 1 (MaSp1) protein's C-terminal (CT) domain, which is crucial for the self-assembly of silk fibers. The researchers discovered that the CT domain can independently form silk-like fibers without the repetitive domain, offering a unique opportunity to gain insights into this process as well as the process of natural spider silk fiber formation. Using techniques such as Fourier transform infrared spectroscopy, X-ray diffraction, and solid-state nuclear magnetic resonance, the study thoroughly examined structure of the macroscopic fibers. The results revealed a helical to β -sheet transition in the fibers. Computational algorithms identified the sequence corresponding to helix No4 as the likely initiator of this structural conversion.

Overall, the study is sound, and the manuscript well written providing high quality results. However, some issues in the study require clearance. I recommend the publication after minor corrections, as follows:

1) Introduction should provide more comprehensive overview on the topic. Herein, the introduction part (lines 47-74) is actually very specifically related to one type of NT, CT and Rep domains used in the research group, although many statements sound general i.e. valid for majority of the studied domains from different species. For example: Lines 56-57. “The NT domain cannot form fibers, neither on its own nor followed by a part of the repetitive...” The literature provides examples where a repetitive domain with a NT domain could be spun into fibers from aqueous solutions. Hence described phenomena is rather related to the specific NT and Rep sequences. Authors should comment on this and extend the part providing more examples on to the issue of fiber formation from recombinant proteins containing NT and/or CTs from different spider species. The source (spider silk species) of the studied or compared NT and CT domains should be noticed in the introduction and whole manuscript for clearness.

Response: Following the reviewer’s recommendation, we have expanded the introduction to broaden the scope and included other types of recombinant spidroins and artificial spinning methods, see page 2. The reviewer correctly points out that there are reports of NT forming fibrils and hydrogels, however those studies that we found used much higher protein concentrations and heat incubation (e.g. 500 mg/ml and 60 °C in Arndt, T. et al. 2022. *Nature Communications* 13:4695). We reiterate that the NT domain to the spidroin used herein (MaSp1 *E. Australis*), neither alone nor in combination with repeats, cannot self-assembly spontaneously (i.e. without

assistance of heat, or artificial spinning methods that uses aqueous organic solvents) to form naked eye-visible silk-like macroscopic fibers. We have clarified that with the passage describing CT domain silk formation, see page 3.

We have read through the text and made changes to make sure that it is always clear what is the source of the mentioned proteins. In our study, the MaSp1 from *E. australis* is the unique source of our recombinant silk.

2) Lines 86-91: Please make clear on what type of material the analyses have been made i.e. natural spider silk fibers or those made of recombinant spider silk proteins.

Response: We have now rewritten the section disclosing each method with examples of recombinant and natural silk experiments according to referred literature, see page 3-4.

3) Line 101: the term “fiber format” in whole manuscript sounds confusing. I recommend using fiber(s) fiber (-like) morphology or fiber form.

Response: We have now changed “*fiber format*” to “*fiber form*” throughout the manuscript as suggested by reviewer.

4) In Figure 2 the Amid I band of the CT domain before transformation (in solution) and after the transformation (in the fibers) should be presented.

Response: For the assessment of overall structure of the recombinant silk proteins before transition to fiber we rely on CD experiments since the experiments are more straightforward to perform in a water solution. Modelling and previous studies confirm the α -helical structure in the soluble protein (As shown in *Rat et al. 2018. Nature Communications 9: 4779; Hedhammar et al. 2008. Biochemistry 47, 3407–3417*).

In addition, we have now added data (Figure S6 and Table S2b) from an experiment when our recombinant CT in solution was placed over the diamond surface of the ATR unit and left to dry, i.e. the sample was not processed with the cyclic expansion/compression and therefore retains its dominant α -helix content, see page 7-8. Taken together, all previous and new experimental data suggest that the CT domain is α -helical in solution.

5) The deconvolution of the Amid I band is confusing. 2nd derivative shows presence of 4 possible peaks (according to the shifts possibly assigned to parallel beta-sheets, alfa-helices, turns and antiparallel beta-sheets (from left to the right), hence it would be more reliable to fit those 4 peaks into the envelope the amid I and afterwards assign the structure according to data from literature (the literature source for the assignment is missing e.g., in Table S6).

Response: Our fitting approach is different from the usual one in that it simultaneously fits the absorbance spectrum and its second derivative (termed co-fitting in the following). Fitting both

spectra takes full advantage of the available spectral information. The usual approach is to use the second derivative spectrum to determine band positions and to use these band positions as starting values for the fit of only the absorbance spectrum. The second derivative of such a fit often does not well reproduce the second derivative of the experimental spectrum. We show below a fit obtained in such a way.

Fig. 1. Absorbance (left) and second derivative (right) spectra of the CT fiber. Red: experimental spectra, black: fit and second derivative of the fit, gray: fitted component bands. The unit of the horizontal axis is wavenumber in units of cm^{-1} .

A four-band fit reproduces the absorbance spectrum well and also the α -helix band in the second derivative spectrum. However, there are clear deviations between the second derivative of the fit and of the experimental spectrum in other regions. In particular, the fitted β -sheet band is too broad as indicated by a too small signal at the peak position and larger signals on both flanks of the band. These deviations can only be avoided by including further bands in the fit model and by using co-fitting. For example, making the β -sheet band narrower, as required by the second derivative spectrum, reduces the absorbance on both sides of this band so that the fit no longer matches the absorbance spectrum. Therefore, new bands on both sides of the 1620 cm^{-1} β -sheet band needed to be introduced. This approach results in the close to perfect agreement between fit and experiment in Figures 2b and 2c of the manuscript. In our evaluation, we always start with the minimum number of bands inferred from the second derivative spectrum. When co-fitting did not produce a satisfactory fit of spectra, we increased the number of component bands.

References to the spectral assignment methods as well as descriptions of the spectral ranges for different classes of secondary structures are now included in the M&M section, page 23.

The spectral ranges used for the assignment of secondary structures: β -sheets $1610\text{--}1641 \text{ cm}^{-1}$, anti-parallel β -sheets additionally $\sim 1690 \text{ cm}^{-1}$, α -helices $1648\text{--}1657 \text{ cm}^{-1}$, and other structures $1642\text{--}1686 \text{ cm}^{-1}$ (i.e. all other secondary structures: irregular, turns, bends, other helix types).

Secondary structures other than α -helices or β -sheets are expected to produce broad bands, which are relatively weak in second derivative spectra. Therefore the 1650 cm^{-1} band was assigned to α -helices. This has now been clarified in the M&M section, page 23.

6) Currently, the appearance of the peaks in the Figure 2C (4 for beta-sheets, 2 for “other” and 1 for α -helices) is unclear. The correct deconvolution-based evaluation of the spectra is however suggested in Figure S3, hence the same should be used in the Figure 2C.

Response: We used a different approach for Figure S7 than for Figure 2 in order to evaluate the β -sheet organizational index in the same way as in the original publication (*Celej, et. al. 2012 Biochemical Journal 443, 719-726*). This has now been clarified in the text, page 8, and the M&M section page 23. However, as outlined above, we do not consider such an approach as optimal for fitting because it does not consider all available spectral information. Deconvoluted spectra as well as second derivative spectra emphasize narrow bands, which can be quite weak in absorbance spectra, while broad bands have a strong effect on absorbance spectra. Therefore, co-fitting considers narrow and broad bands in the same procedure.

7) Figure 3, what kind of steric zipper structure are shown in Figure 4C and D, an arbitrary or particular ones?

Response: The steric zipper structures were only intended as illustrations. We decided to remove them to avoid any misleading interpretation.

The CT domain is apparently stable at different pH as the CD spectra in Figure S1 clearly show α -helical structure in the pH range 12.8-4.8, whereas the hydrophobic peptide from the Helix Nr. 4 readily transforms into beta-sheets, hence the relation of this experiments in Figure 6A to the explanation of the observed CT transformation is rather elusive, especially because the pH shift plays no role in the experiment presented to the CT and CT-Rep transformation into fiber. Please clear this. How stable is the CT domain and CT-Rep at pH range 7.5-4.5 (thermal stability at different pH)? Also, for the demonstration of the differences in the pH triggered transformation, Figure S1 should be included into Figure 6, whereas Figure 6B could be excluded in supporting information.

Response: CT is less thermo stable at acidic pH than at neutral pH. As shown in *Rat et al.2018. (Nature Communications volume 9, Article number: 4779)*, CT is sensitive to pH shifts. The effect of pH on the structure of the recombinant CT domain of *Nephila clavipes* dragline silk protein has also been depicted by *Hagn et al. 2010 (Nature 465, 239-U131)*. However, pH-induced structural transition is slower in full-length CT than for the helix 4 peptide and often requires extra stimuli to form fibers. We have now included CD data showing the lower structural stability of CT at low pH in the SI (Figure S1).

The pH is not the only variable that regulates the fiber formation of spidroins. The recombinant spidroins of this study (*E. australis* 4RepCT and CT) can, for example, form fibers independently of pH changes when exposed to shear-stress. This is the approach we use for fiber formation in this work. This has now been clarified in the result section, page 6, and the M&M section page 22.

The beta-sheet rich structure in the CT fibers as well as those the peptide fibrils are proposed to be amyloid like. Hence experiments typical for amyloids such as ThT binding onto fibrils or “green birefringence” of Congo red on the fibers should be demonstrated.

Response: We have now included data from experiments using the dye Amytracker (Ebba Biotech AB), that is a fluorescent optotracer for detection of amyloid proteins and repetitive arrangement of β -sheets. As seen from Figure S9a, fluorescence microscopy confirms binding of the Amytracker 630 dye to helix 4 peptide fibrils, which further supports the presence of amyloid-like β -sheet structure. Moreover, the combination of β -sheet content (revealed by CD) and the appearance of nanofibrils strongly suggest an amyloid-like arrangement of the peptide. It is hard to imagine any other fibrillar β -sheet structures formed by a short peptide (and if such structure exists, it would most likely also bind common dyes used for amyloid detection).

In the case of the CT fibers, FTIR analysis show that they have a high β -sheet content and X-ray diffraction patterns that are in agreement with an amyloid-like packing. Moreover, we have now included data from staining with Amytracker 680 dye, further supporting presence of repetitive arrangement of β -sheets, see Figure S9 b. This is now also clarified in the text, page 11 and page 18.

Reviewer #2.

De Oliveira present an interesting study that provides a valuable expansion to knowledge of spider silk fibre formation. Namely, the role of the nonrepetitive C-terminal domain (CT) in self-assembly is probed in major ampullate-silk based constructs, with focus on the CT itself from *E. australis*. A series of analyses of this CT are presented in the solid-state (ATR-FTIR, SRD, solid-state NMR) with underpinning by computational analyses (homology modeling, amyloid propensity prediction, evaluation of modeled peptide energy by Rosetta). In some instances, direct comparison is carried out with the partial spidroin construct 4RepCT, which is composed of four copies of poly-Ala/Gly stretches from the repetitive domain fused to the CT.

The rationale for the study herein is that the CT has been shown to self-assemble previously into fibres (albeit with lower stability than 4RepCT), providing a means to evaluate the putative involvement of the CT in triggering of fibre formation through a role in alignment of the repetitive domains. The authors convincingly demonstrate that CT-based fibres have mixed α -helical and β -sheet composition and show that the β -sheet components are distinct from those of the Rep in terms of the observed inter-sheet spacing, consistent with residues having longer sidechains. Based upon chemical shift evaluation in solid-state NMR, the structural transition is linked to helix 4 of the CT, a segment of the CT that – based upon homology modeling – would be expected to form an interface with helix 4 of the second protomer in a dimer where the helix 4’s of each protomer are aligned in a parallel fashion (at least in instances where this is disulfide linked). There are definitely some sections which would benefit from a little bit more proofreading and polishing,

but as a whole this is both a well-written and easy to follow manuscript. Notwithstanding my generally positive assessment of the work and impression that this is both of broad interest and valuable to the silk field more specifically, I do have some concerns that I feel should be addressed prior to consideration for publication. Proofread text.

Thanks for the positive comments! We have now proofread the text, and checked with Grammarly, several times before resubmission.

1) First off – most importantly – with all of the focus on the CT dimer state and potential for the (putatively) disulfide linked helix 4 to influence self-assembly, there appears to be a major missing set of information with respect to evaluation of the state of the Cys residues in the full-length CT and synthetic peptide. The authors appear to be presuming there is a disulfide (e.g., Figure 7), but is there any experimental evidence to demonstrate whether the CT being used in a given sample/condition is in a disulfide linked dimeric state (or not)? I may have missed this, but I do not see any mention of this anywhere or experimental demonstration of this. Clearly, this has important ramifications for all of the interpretation. Related to this (but independent of demonstration of dimer formation and stability), for sample IV (mixed ^{13}C -labelled + ^{15}N -labelled protein), more detail would be beneficial with respect to (1) oxidation state and (2) when proteins were allowed to disulfide link (i.e., would mixed $^{13}\text{C}/^{15}\text{N}$ -labelled disulfide linked dimers be expected at statistical prevalence, or was dimer formation carried out prior to mixing of the proteins?) In general, some additional detail about the state of the protein in terms of the disulfide is needed in all sample preparations. Likewise, while the CT peptide is discussed in terms of disulfide formation, there is no mention of this peptide being prepared in the oxidized state and/or oxidized prior to usage to form disulfide-linked dimers, nor is it shown to be a dimer by any experimental methodology. On page 17 (point iv, lines 357-359), can further comment/discussion be made about the ability of the S-S linkage to be accommodated in the sheet (it's noted that the disulfide provides a favourable situation for beta-sheet formation, but disulfides actually tend to perturb at least one strand in a sheet and are relatively rarely observed in beta-sheets – the study by Almeida et al. (2012, JACS, 134: 75) may be beneficial to consider as part of this expanded rationalization?) Finally, one related note is that Cys isn't assigned in the solid-state NMR data – could the unassigned cross-peak at ~38 ppm / 58 ppm be Cys (assuming independent evidence of disulfide linking, before using this cross-peak as evidence of the cystine state) and if so what would the inferred secondary structure be? (I recognize this is only a single residue, so perhaps the signal-to-noise is not sufficient to pick this up.)

Response: Previous studies (using native Size Exclusion Chromatography) have shown that the same soluble CT domain protein is a dimer directly after expression in *E. coli* and native purification, without any treatment for oxidation (Hedhammar et al. Structural properties of recombinant nonrepetitive and repetitive parts of major ampullate spidroin 1 from *Euprosthenois australis*: implications for fiber formation. *Biochemistry* **47**, 3407-17 [2008].) In fact, during those studies we were not able to find any monomeric form of CT under any tested native conditions, not even for a Cys-Ser mutant, indicating that the dimer is stabilized by other means. This is in agreement with previous NMR spectroscopy analyses, as shown in Hagn et al. *A conserved spider silk domain acts as a molecular switch that controls fibre assembly*, *Nature* **465**, 239-U131 (2010), and the structure which is connected by disulfide bond in PDB ID: 2KHM. The region with the

Cys residue contains a hydrophobic stretch, shown in *Rat et al. Two-step self-assembly of a spider silk molecular clamp Nature Communications volume 9, Article number: 4779 (2018)*, which is proposed to drive the first steps of the hydrophobic collapse in a two-step folding pathway.

Our buffer conditions for purification and fiber formation, and analyses are without any reducing agents, only 20 mM Tris, pH 8. Since the dimeric structure is stable, we did not take any special action to ensure that the Cys-residues are in the oxidized state. With that said, we have now included an SDS-PAGE analysis of the denatured soluble CT domain in reducing and non-reducing conditions (Figure S3). The gel under reducing conditions presents one band for the monomeric CT domain around 10 kDa while the non-reducing conditions presents two bands, one around 10 kDa and another around 20 kDa that suggests the presence of the dimer when dithiothreitol (DTT) is not used in the analysis. These results confirm that CT domain is (at least partially) present as a disulfide-linked dimer in the sample of the soluble CT domain. We also analyzed the size of the building blocks in the CT fibers. The CT fibers were decomposed in 6 M urea and analyzed by SDS-PAGE without reducing agent. On that gel (Figure S4), only the band corresponding to a dimer (20 kDa) is seen. Hence, we conclude that CT in the fiber state primarily exists as a disulfide linked dimer. This is now described in the result section, page 4-5.

For the synthetic peptide, we have MALDI-TOF data of the peptide H4 (Figure S14) that confirms the presence of monomers as well as dimers with sizes around 3 kDa and 6 kDa, respectively. This is now described in the result section, page 16.

For the mixed ¹³C/¹⁵N-labelled NMR sample, the purified proteins were mixed before fiber formation. Hence, the soluble dimers are expected to exist in the sample before fiber assembly. As we cannot follow the refolding of CT from a single polypeptide chain, it is not possible to say if the PAIN experiment primarily reports N-C contacts between pre-formed dimer units or if mixing also occurs on a single chain level.

We thank the reviewer for highlighting the work of Almeida et al. (2012, *JACS*, 134: 75). That study shows that although disulfides may destabilize β -sheets they could also stabilize them if the bond is located at the edge of the β -strands. In CT, the Cys residue is located at the edge of helix 4 and may as well be located at the edge of a β -strand (according to figure 5). Hence, it could function as a “gatekeeper” in the structural transition. In the synthetic peptide, Cys is in the beginning of the chain and would end up at the edge of the formed β -sheet.

We did not find enough spectral evidence to assign the mentioned NMR resonance to Cys. If the peak does originate from Cys, it would be compatible with the oxidized state in α -helical conformation.

2) It is a bit unclear how the Consurf analysis described in the methods was used and what the source of the 143 unique sequences noted as being used for “calculation” (what sort of calculation?) were vs. the 217 “hits” that were noted. Was this used as part of the homology modeling with SwissModel, or is that strictly homology model? What kind of quality metrics/statistics are available for the homology model (over and above the reported % identity)? Note in methods (line 568): “determined” should be determined.

Response: The Consurf and SwissModel were independent analyses using the same protein sequence.

For the Consurf, the automated pipeline of sequence analysis is described as follows: the protein entry sequence is BLASTed against the UNIREF-90 database (UniProt). 217 hits were found and after the redundant homologous sequences had been removed, we ended up with 143 unique sequences. The algorithm performs a multiple alignment assignment (MSA) and generates the phylogenetic tree. The program computes the position-specific evolutionary rates with an empirical Bayesian method and depicts each amino acid in 9 different conservation degrees. The materials and methods section has now been re-written for better comprehension and extra references added, see page 27.

For the SWISS-MODEL homology model, the general cut-off for sequence identity is 30-35%, (Xiang, Z., 2006. *Curr. Protein Pept. Sci.* 7(3):217-27). The acceptable threshold of 35% sequence identity is fulfilled with the protein model of CT domain of *E.australis* presenting a sequence identity of 60.4%.

The quality metrics/statistics are available from the SWISS-MODEL server, including the Ramachandran plots and MolProbity. The model presented for the CT domain of *E.australis* has no residue in disallowed regions and over 95% in allowed or generous allowed regions, providing a good resolution. The MolProbity Score combined protein quality score that reflects the crystallographic resolution at which such a quality would be expected. The SWISS-MODEL structure assessment page runs MolProbity version 4.4. (Richardson *et al.* 2007. *Nucl. Acids. Res.* 35: W375-83). The MolProbity of the CT domain model of *E.australis* 1.78 is considered appropriated and acceptable.

The method description has been updated (page 27). The passage in methods (Page 27) is corrected to determined.

3) It would be great to have a more direct overlay of the 2nd derivative and absorbance data if Figure 2 (or at least some guide-lines to allow for comparative assessment of the components noted in each.)

Response: Thanks for the suggestion. We have now included guide-lines linking the peaks of secondary derivative peaks to absorbance in figure 2.

4) Although an optical micrograph of the CT-derived fibres is presented in Figure 2 and an SEM micrograph is presented Figure 7, details of the fibre morphology and features (as well as of the microscopy methodology) are not provided. Are these truly “silk-like” fibres that recapitulate previous studies of the CT?

Response: The macroscopic fibers, made by self-assembly of the CT-domain, investigated here resemble silk-like fibers of previous studies with similar coloration, and longitudinal size. The CT domain fibers are more brittle and easier to break mechanically (limited moldability, plasticity) when compared to those made using same self-assembly method of the recombinant 4RepCT. This has also been reported in previous studies (Structural properties of recombinant nonrepetitive and repetitive parts of major ampullate spidroin 1 from *Euprosthenops australis*: implications for fiber

formation. Hedhammar *et al.* Biochemistry 2008, 47, 3407–3417, and Interfacial Behavior of Recombinant Spider Silk Protein Parts Reveals Cues on the Silk Assembly Mechanism Nilebäck *et al.*, Langmuir 2018, 34, 11795–11805).

We have included a new photo of fibers from the CT domain (Figure S4a) and SEM image (Figure S5). The equipment used for Scanning electron microscopy is Gemini Ultra 55, Zeiss, Germany, described in the Materials & Methods section, page 28.

5) p. 9, line 177 – presumably this should be a reference to Figure 1 not Figure 4?

Response: The intention here was to point out the secondary structure state of Glu. We realized that it was confusing and not necessary, so we decided to remove the reference to the figure.

6) p. 9, para. 2 – is it possible to show a more direct comparison to Rep domain-based fibre diffraction (or, lacking this, wildtype fibre diffraction where the repetitive domain would presumably be dominating) in terms of backing up the statement that 4RepCT is a superposition [convolution may be better] of the CT domain fibre and the β -poly-(L-Ala) structure? From the data provided in Fig. 3 alone, this statement is hard to assess.

Response: We do have X-ray diffraction data of a sample with the 4Rep protein (i.e. four times repeated poly-alanine glycine rich motives) of MaSp1 from *E. australis* produced and purified using similar protocols. However, since this construct forms aggregates rather than fibers (i.e. no order) (Hedhammar, M., *et al.* 2008. Biochemistry 47, 3407-3417) we did not include it in the first version of the manuscript. Based on the reviewer's comment, we have now decided to include the data in the supplementary materials (Figure S8c and S8d). The diffraction pattern facilitates the understanding of convolution between 4Rep and CT domain of MaSp1 *E. australis*, clarified in the text, page 10, line 219-224. Note that the 4Rep sample does not present the larger inter-strand distances (9.7 Å) seen in CT and 4RepCT but do have the dominant diffraction assigned to poly-alanine-like crystallites (Table S3).

7) p. 10, lines 205-207 – it is a bit unclear why the aromatic content would be “lower” in 4RepCT than in a natural silk, at least if derived from the same protein. (i.e., per unit mass in the MAS rotor, the proportion of aromatics should be ~similar, shouldn't it?)

Response: The aromatic (Tyr) signals mainly come from the repetitive part and would be more prominent in natural silk that has more extensive content of repetitive regions in relation to the terminal domains. This is now clarified in the text, page 11.

8) p. 10, para 2 – for arguments about relative abundances etc., it would be easier to follow and assess this if the authors present a table or histogram of amino acid #'s/proportions [could be supplementary info] in the Rep vs. CT domains.

Response: The histogram of amino acid #'s proportions is now included with supplementary information (Figure S11).

9) p. 10, Lines 221-222 – the inference is made that there is disorder in the fibre state due to the fact that ^{15}N spectra were poorly resolved. This seems like a large leap of logic, given that there could be a variety of causes of broadening and loss of resolution over and above disorder.

Response: We did not intend to claim that the protein chains are disordered. Rather the intention was to say that the sample may contain a mixture protein chains in different types of structures. We have corrected the text accordingly, see page 12.

10) p. 10, lines 224-226 and subsequent to this – there is no detail provided about how chemical shifts were evaluated with respect to secondary structure inferences, nor about the “robustness” of a given assignment. I am also concerned about the ^{13}C referencing noted, where TMS was noted to be the 0 ppm reference – biomolecular secondary structure evaluation is typically evaluated with respect to DSS at 0 ppm, which is in turn substantially offset from TMS for ^{13}C , so if the data presented are indeed referenced to TMS then this would lead to a consistent offset in chemical shifts that would bias/modify secondary structure inferences.

Response: Assignments were done using the semi-automated assignment system in CCPNMR software (*Vranken, W.F., et al. 2005. Proteins-Structure Function and Bioinformatics 59, 687-696*) that identifies amino acids and suggests assignments based on probabilities of the internal database. The assignment of amino acid residues relies on the identification of cross-peaks between all side chain atoms. This means that the assignment for amino acids with distinct chemical shifts in their side chains (e.g. Ala, Ile, Val) will be more “robust” (higher confidence) than those without such shifts (e.g. Glu, Gln, Asn, Asp). All reported shifts were validated in the full set of recorded spectra. Assignments with apparent inconsistencies were excluded from the final set of reported assignments. Interpretations of chemical shifts in terms of secondary structure were based on the “Wishart” protein secondary structure parameters (*Wishart, D.S. 2011. Prog. Nucl. Magn. Reson. 58: 62–87*). We have made some clarifications about the assignment procedure in the methods section (page 26).

Regarding the ^{13}C referencing, we realized that there was an error in the methods text. The shifts were corrected by the TMS - DSS difference (+2.67 ppm) and the reported values are relative to the DSS reference. We have corrected the description in the M&M section (page 26).

11) p. 10, lines 227-229 – what is the rationale that “additional residues” are observed through longer (or different) DARR mixing times? I would expect further spin diffusion with a longer mix (i.e., observing dipolar species that are further separated) but not necessarily the ability to see “additional” residues.

Response: What is meant is that longer mixing times produced stronger cross-peaks between nuclei “far” apart within the same amino acid, which allowed a more robust assignment. We have rephrased this sentence, page 12.

12) p. 12, Figure 4 – fairly major conclusions are drawn from the Gln/Glu assignments that are made (e.g., p. 16, lines 332-333). However, these appear to be based on a single (relatively weak) cross-peak for each residue type. One might also argue that there's possibility that one of the CB shifts is actually a CG shift, for example, in that perhaps only one of these amino acid types is observed? This comes back to point # 10 about the method of assignment and robustness of assignments

Response: The assignment of Gln/Glu (in total 7 positions in CT) is based on cross-peaks involving C α , C β and C γ . In particular, the observation of cross-peaks involving C γ supports their presence in the spectrum. We cannot distinguish between Glu and Gln, the chemical shift could be one or the other or both. The specific shifts (C β and C γ) for Glu and Gln, were assigned based on database statistics. The assigned shifts are:

Residue	C α	C β	C γ
Glu	53.6	33.1	35.6
Gln	53.6	31.5	34.8

Alternative assignments include Cys, Met, Arg. The C α and C β shifts are compatible with Cys (Reduce form), with typical shifts C α ~ 57 ppm and C β ~30 ppm) and we found our CT domain fiber in oxidized form (Figure S4). However, the assignment to Cys does not explain the presence of the peak at 35 ppm (C γ). Assignment to Met (3 positions in CT) is consistent with C α (~54 ppm) and C β (~35 ppm) and C γ (~32. ppm). However, we do not observe crosspeaks to C ϵ (~20 ppm). Assignment to Arg (2 positions in CT) is consistent with C α (~55 ppm) and C β (~32 ppm), but we do not find the expected crosspeaks for C δ (~43 ppm), C γ (~27 ppm) and C ζ (~160 ppm).

Taken together, the assignment of Glu/Gln is the best match for these cross-peaks and these residues are also more prevalent in the sequence than the alternative. We have added a short section in the Methods part about the robustness of the assignment (see response to issue 10, and page 26).

13) p. 13, line 258 – would “undergoes a structural conversion” be better worded as “undergoes a partial structural conversion”? This appears to be the main conclusion?

Response: Our data shows that the structure changes upon fiber formation. However, we cannot determine to what extent this is a partial conversion of every protein molecule or heterogeneous conversion among molecules. Hence, we find the more general statement “undergoes a structural conversion” to be more accurate. A clarification has been introduced on page 12.

14) p. 14, Figure 5 and related discussion – the inclusion of Rosetta energies seems out of place (until one realizes this is output from ZipperDB) – some additional clarification as to the methodology used and information obtained would be beneficial. It may also be helpful to have

the color scale legend used for Rosetta energy scaled in such a way that the length of the arrow matches the numerical values on the y-axis to a given color or else to present this differently so that the coloured arrow is not directly aligned with the y-axis.

Response: We have now in Figure 5 clarified that the Rosetta energy is indeed an output of ZipperDB and simplified the graph with two color scheme, red below the threshold and yellow above the threshold. The color scheme of the y-axis arrow is also aligned now.

15) p. 15, lines 297-298 – the noted behaviour was not entirely clear on first read. I would suggest rewording this.

Response: The sentence has been rewritten to say: “Interestingly, the sample of *helix N^o4* peptide at pH 8 also shifted minima toward a single peak around 220 nm (β -sheet pattern) when kept for extended incubation time (16-20 h, at room temperature) (Figure S15a).”, see page 16.

16) p. 16, para. 1 – there’s noted to be a pH-dependent trend with respect to a progressive β -sheet transformation. Is there a [clear] pH-dependence of the kinetics of this transformation? (It’s not entirely clear what type of pH-dependent behaviour was observed.) It’s also not clear why the data at high pH are not shown?

Response: The presented data shows that there is a pH-dependent trend with respect to a progressive β -sheet transformation. The measurements were done immediately after solubilization and after 16-20 h. We have not performed a full kinetic assay for the structural conversion of the peptide. This will be the focus of a planned follow-up study. The CD data for high pH (11.2 and 11.4) is now included in Figure 6a.

17) p. 16, para. 2 – the AFM data certainly appear to show fibrillar species, but there is a concern that these micrographs show imaging issues (e.g., loss of contact) and have height scales that start from substantial negative values. It’s not clear in the methods that these experiments were repeated, nor is it clear that these are “representative” images as opposed to a single pair of images collected for a single fibre at two resolutions. All in all, as presented, the AFM data are not highly compelling.

Response: The presented images are indeed representative for the type of nanofibrils found in the peptide samples after decreasing the pH. This is now emphasized in the text (page 16, legend of Fig. 6) and AFM images from a second sample at a slightly different pH (pH 2 instead of pH 3) have been added in the supporting information (Figure S15).

18) p. 20, para. 1 – the expression and purification protocols are somewhat sparse. (What form of fusion protein was used? How was cleavage carried out? What level of purity was achieved [i.e., as demonstrated by SDS-PAGE, for example]?) Perhaps this is all exactly as described in ref. #6, but the level of detail is still not sufficient and the protein purity following cleavage is important to document and demonstrate.

Response: For the expression of CT domain from MaSp1 of *E. australis* in *E. coli* we used His-tag with thioredoxin as solubility partner, the tag was cleaved off by protease and the CT domain was purified, as described in previous work (Stark, M., et al. 2007. *Biomacromolecules* 8, 1695-1701). We added a brief description in the M&M section (page 21).

We have now included the SDS-PAGE gel of this purified CT domain (Figure S3).

19) p. 22, lines 488-490 – the statement that the two constructs have no major differences in diffraction pattern is a bit hard to assess in direct comparison of Figure S8 and Figure 3, where the annotated features are not always visible in Figure S8 and with the differences in presentation with respect to q values plotted and apparent features that are visible in Figure S8(c) that differ from those in Figure 3.

Response: The purpose of the additional X-ray diffraction data (Figure S16) was to corroborate the results from the synchrotron experiments as they were carried out on samples with a small sequence difference. The experiments from the SI were conducted in tabletop diffraction instrument, and the quality of the data is indeed not as high as for the data in the main text. Nevertheless, we do not observe any major discrepancies, which support the claim that the interpretation of the synchrotron data is valid also for the protein construct investigated with the other techniques.

20) pp.22-23 – it comes as a bit of a surprise in the methods that the fibres studied by solid-state NMR are “hydrated” (p. 23). This should be more clearly noted in the results/discussion in terms of clarification of the manner in which the solid-state NMR samples are prepared.

Response: The hydration procedure means that we add a few microliters of water to the dry NMR sample in order to get sharper lines. This is a standard procedure for high resolution solid-state NMR which we also used in previous work (e.g. Lendel et al. 2014. *Angew. Chem. Int. Ed.* 53, 12756–12760). This is now clarified in the M&M section, page 24.

21) Para. spanning pp. 23-24 – there’s a lot of detail provided on backbone assignments when it seems this process wasn’t actually carried out. This could perhaps be shortened or removed as it wasn’t applied?

Response: This section concerns the ^{15}N - ^{13}C correlation spectra. To clarify this, we have changed ‘backbone assignments’ to “ ^{15}N assignments”.

Reviewer #3.

The work reports that C-terminal domain in major ampullate spidroin 1 is capable to form fibrillar structures without the presence of MaS1 repetitive domains. Overall, this is a highly rigorous and useful paper that uses an ensemble of spectroscopy methods, including FTIR, CD, NMR, X-ray

for analysis of CT assembly. I believe there are extra minor details that could be added to improve the clarity of the paper and the certainty of the interpretations.

1) Authors have modeled overall folding of CT domain and confirmed this experimentally with circular dichroism analysis (Figure S1). However, Fig. S1 depicts only original spectra with no data interpretation. I suggest to add chart showing the fraction of beta-sheet, random coil/helical content based on depicted CD analysis. Also, how authors explain that CD spectra of CT at pH 4.8 and pH 12.8 almost perfectly overlap (at least for my unequipped eye)? How relevant those extreme pH values to actual pH gradient in spider silk gland?

Response: We have now included a decomposition of the CD data shown in figure 1, see Table S1. We used the software BeStSel (*Micsonai et al. 2022, Nucleic Acids Res. 50(W1): W90-W98*) to extract quantitative information about the secondary structure content.

Although the soluble state of CT initially retains its secondary structure when exposed to low pH, the structural conversion occurs faster than at higher pH values. Acidic pH disturbs the stabilizing salt bridges in CT and facilitates the structural transition, a process that is further enhanced additional stimuli such as shear-stress.

A pH value around 5 is found close to the end stage of spiders spinning channels. The alkaline pH is mainly used in this paper to solubilize the peptide. The CD data at different pH values is included to demonstrate that the pH does affect the initial structural stability of the CT domain, see Figure S1.

2) If I understood correctly the fiber has been formed via shaking (stretch-relaxation cycles method)? If not please expand experimental details. If yes, can authors provide shaking conditions in the materials and methods section.

Response: Yes, it is the stretch-relaxation cycles method. The method has been described before (Kvick, M. et al., Cyclic Expansion/Compression of the Air-Liquid Interface as a Simple Method to Produce Silk Fibers. *Macromol Biosci* **21**, e2000227 [2021]). We now provide additional information about amplitude and speed of the rocking table in the M&M section (page 22).

3) FTIR fitting... Can authors provide ranges for beta-sheet, helices peaks assignment. Also, how “other” secondary structure has been interpreted. Has it been assigned to helical conformation or to beta-sheet? “Other” conformation constitute 15.55% of the overall structure and can affect data interpretation. The sample of FTIR (ATR) analysis contained only fibrillated peptide or mixture of fibrillated and monomeric peptide? There are some useful papers describing FTIR analysis of the silks that I recommend to cite if not cited yet (1) *Nat Commun* **13**, 7856 (2022). <https://doi.org/10.1038/s4146702235505w>; 2) <https://doi.org/10.1002/app.21346>; 3) <http://doi.org/10.1016/j.vibspec.2007.10.003>)

Response: The spectral ranges used for the assignment of secondary structures were:

β -sheets: 1610–1641 cm^{-1} ; anti-parallel β -sheets additionally $\sim 1690 \text{ cm}^{-1}$; α -helices 1648–1657 cm^{-1} ; and other structures 1642–1686 cm^{-1} (i.e. all other secondary structures, irregular, turns, bends, other helix types). Secondary structures other than α -helices or β -sheets are expected to produce broad bands, which are relatively weak in second derivative spectra. Therefore the 1650 cm^{-1} band was assigned to α -helices. The materials and methods of FTIR has been updated, page 23.

4) Can author comment on the relative ratio between parallel and anti-parallel beta-sheet content.

Response: Our value for the β -sheet organizational index is 0.1, which is between those for proteins rich in *antiparallel* β -sheets (0.2-0.3) and for proteins rich in *parallel* β -sheets (≤ 0.05). From these values, it can be concluded that the relative content of β -sheets with antiparallel strand orientation is approximately between 15% (using 0.05 for 100% parallel sheets and 0.3 for 100% antiparallel sheets) and 50% (using 0 for 100% parallel sheets and 0.2 for 100% antiparallel sheets). However, the index has not been correlated with the content of mixtures of antiparallel and parallel sheets, so we rather refrain from stating numbers which will have a rather large error. We have now stated in the text, page 8, that we have a mixture of parallel and antiparallel β -sheets with some prevalence for parallel sheets, which is a fair description of what the numbers indicate.

5) Experimental section on CD data interpretation/fitting/analysis is missing.

Response: We have now included a new paragraph with CD data interpretation/fitting/analysis, at the beginning of the results section (page 4). The BeStSel analysis is also mentioned in the CD methods section (page 26).

6) Again how chosen pH values for peptide fibrillation are relevant to physiological conditions of spider? Author should comment on this in the main text.

Response: The relevant pH range in for spider spinning apparatus is between 7.2 to 5.7 (*Andersson et. al., 2014. PLoS Biol 12, e1001921*), which is stated on page 19. We here explored the effect of pH within a wider range (pH 2-12) to get a better comprehension of the phenomena. The pH range of interest for the natural spider silk is included. Additional information about the pH dependence may be important in developing new silk materials and artificial spinning.

7) X-ray data: Out of curiosity I am wondering whether authors succeeded to detect the pattern of H-bond network? (see relevant work: Commun Chem 4,62(2021). <https://doi.org/10.1038/s42004-021-00494-2>)

Response: The suggested relevant work uses XPS to investigate the patterns of the H-bond networks in nanofibrils formed by various designed peptide variants. We find the approach interesting, but we have not measured any XPS data for our samples. Considering that such experiments would require substantial work to set up and optimize the measurements, we consider that to be a goal of future work.

8) What was the solvent for the peptide solubilization? How pH has been adjusted?

Response: The buffer used for peptide solubilization was 20 mM Tris pH 12. The pH was then adjusted to final pH values by drop-wise addition of 20 mM Tris adjusted in pH gradient from 2 and 12. This is now described in the M&M section, page 26.

Reviewers' Comments:

Reviewer #1:

Remarks to the Author:

My concerns have been addressed in the revisions and I recommend the manuscript for publication.

Reviewer #2:

Remarks to the Author:

All of my comments and concerns have been comprehensively discussed in the revised manuscript and rebuttal submitted by De Oliveira et al. In my opinion, this manuscript describing a very exciting and substantial body of work is now suitable for publication.

Reviewer #3:

Remarks to the Author:

Authors have addressed my comments. I recommend to accept.